# TODDLERDIFFUSION: INTERACTIVE STRUCTURED IMAGE GENERATION WITH CASCADED SCHRÖDINGER BRIDGE

**Eslam Abdelrahman** [1], **Liangbing Zhao** [1], **Vincent Tao Hu** [2],
**Matthieu Cord** [3], **Patrick Perez** [4], **Mohamed Elhoseiny** [1]
[1]KAUST    [2]LMU    [3] Valeo AI    [4] Kyutai

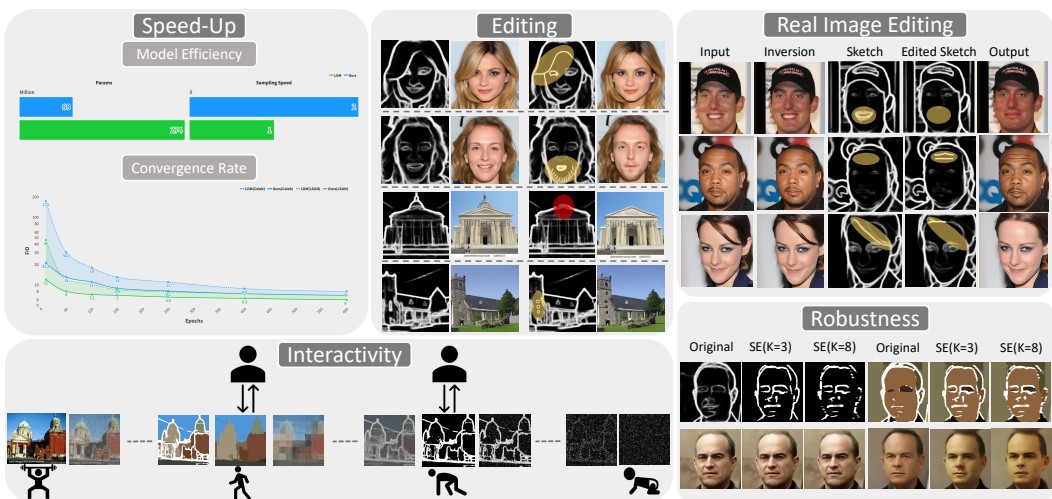

Figure 1: An overview of ToddlerDiffusion capabilities: 1) Speed-Up: Performing on the bar compared to LDM while being 2× faster, using 3× smaller architecture. 2) Interactivity: Producing intermediate outputs, e.g., sketch and palette. 3) Editing: Offering free, robust editing capabilities even for real images. 4) Robustness: Our method is robust against input condition perturbations, e.g., sketch sparsity.

## ABSTRACT

Diffusion models break down the challenging task of generating data from high-dimensional distributions into a series of easier denoising steps. Inspired by this paradigm, we propose a novel approach that extends the diffusion framework into modality space, decomposing the complex task of RGB image generation into simpler, interpretable stages. Our method, termed ToddlerDiffusion, cascades modality-specific models, each responsible for generating an intermediate representation, such as contours, palettes, and detailed textures, ultimately culminating in a high-quality RGB image. Instead of relying on the naive LDM concatenation conditioning mechanism to connect the different stages together, we employ Schrödinger Bridge to determine the optimal transport between different modalities. Although employing a cascaded pipeline introduces more stages, which could lead to a more complex architecture, each stage is meticulously formulated for efficiency and accuracy, surpassing Stable-Diffusion (LDM) performance. Modality composition not only enhances overall performance but enables emerging proprieties such as consistent editing, interaction capabilities, high-level interpretability, and faster convergence and sampling rate. Extensive experiments on diverse datasets, including LSUN-Churches, ImageNet, CelebHQ, and LAION-Art, demonstrate the efficacy of our approach, consistently outperforming state-of-the-art methods. For instance, ToddlerDiffusion achieves notable efficiency, matching LDM performance on LSUN-Churches while operating 2× faster with a 3× smaller architecture. The project website is available at: $https://toddlerdiffusion.github.io/website/$

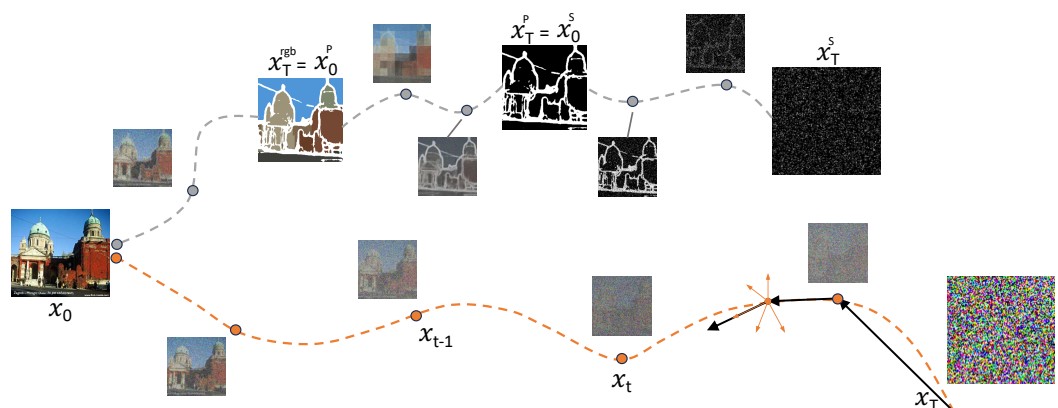

Figure 2: Trajectory comparison. We compare our approach (Top) against LDM (bottom) in terms of trajectory. Instead of starting from pure noise such as LDM, our approach, ToddlerDiffusion, leverages the intermediate stages to achieve a more steady and shorter trajectory.

# 1 INTRODUCTION

In recent years, diffusion models (Sohl-Dickstein et al., 2015; Ho et al., 2020; Song et al., 2020b) have made significant strides across diverse domains, revolutionizing image synthesis and related tasks by transforming noisy, unstructured data into coherent representations through incremental diffusion steps (Ho et al., 2020; Preechakul et al., 2022; Dhariwal & Nichol, 2021; Nichol & Dhariwal, 2021). Their versatility extends beyond image generation to tasks such as image denoising (Gong et al., 2023; Kulikov et al., 2023; Zhu et al., 2023), inpainting (Lugmayr et al., 2022; Alt et al., 2022), super-resolution (Li et al., 2022; Gao et al., 2023), and applications in 3D content creation (Anciukevičius et al., 2023; Qian et al., 2023; Poole et al., 2022), data augmentation (Carlini et al., 2023; Li et al., 2023c; Voetman et al., 2023), medical imaging (Chung et al., 2022; Kazerouni et al., 2022; Wu et al., 2022; Wolleb et al., 2022), anomaly detection, and more. Diffusion models, such as Denoising Diffusion Probabilistic Models (DDPMs) (Ho et al., 2020), draw inspiration from the diffusion process in physics, breaking down the intricate task of generating data from high-dimensional distributions into a sequence of simpler denoising steps. In this framework, instead of generating an image in a single step, the task is reformulated as a gradual denoising process, where a noisy image is iteratively refined until the final, detailed image emerges. This step-by-step approach leverages the power of decomposition: by solving a series of simpler tasks, we can ultimately solve the original, more challenging problem.

Inspired by this philosophy, we propose ToddlerDiffusion, a novel method extending this decomposition principle beyond just noise removal. Instead of tackling the image generation problem in one go, we decompose it into modality-specific stages. Each stage focuses on generating a different aspect or modality of the image, such as abstract contours or sketches, followed by color palettes, and finally, the detailed RGB image. This approach mimics the learning process of a child developing artistic skills (Stevens, 2012; Marr, 2010; 1974), where the child first learns to draw basic outlines, then adds colors, and finally fills in the details. We use the term "ToddlerDiffusion" to reflect this gradual, step-by-step learning and generation process.

Despite the intuitive appeal of task decomposition, it introduces significant challenges. More stages mean more parameters, longer training times, and potentially slower sampling speeds. However, our hypothesis is that by addressing simpler tasks at each stage, we can leverage smaller architectures tailored to each modality, resulting in a more compact overall model compared to traditional single-stage architectures. Moreover, multi-stage frameworks often suffer from error accumulation (Ho et al., 2022; Huang et al., 2023), where inaccuracies in earlier stages can propagate and amplify through subsequent stages. However, we showed that by using simple techniques such as condition truncation dropout, we can significantly mitigate the error accumulation, Appendix A.7 Additionally, learning these intermediate representations, e.g., counters, explicitly or implicitly is challenging, where the first requires paired data and the latter requires special complicated designs (Singh et al., 2019; Li et al., 2021). To tackle the guidance challenge, we leverage pseudo GT and show that this noisy GT is enough to learn robust intermediate modalities.

To implement this framework, we considered two approaches. The first is the standard Latent Diffusion Model (LDM) (Rombach et al., 2022) conditioning mechanism, using techniques such as cross-attention and concatenation to link stages. Given our focus on spatial modalities like sketches, concatenation seemed more appropriate. However, this method proved insufficient for achieving our hypothesis due to its weak conditioning, which results in poor information transfer between stages. We instead adopt a more principled approach by employing the Schrödinger Bridge (Wang et al., 2021; Li et al., 2023a), a method from stochastic optimal transport theory, to determine the optimal path between modalities at each stage. Unlike simple concatenation, which starts each stage from a state of pure noise, the Schrödinger Bridge enables us to begin each stage from a meaningful signal with a higher Signal-to-Noise Ratio (SNR) (Kingma et al., 2021), as illustrated in Figure 2. This leads to more effective and stable progression through the generation process, significantly reducing the denoising steps required during training and testing.

This structured, multi-stage approach not only allows for a more efficient and stable generation process but also enables a range of emerging capabilities. ToddlerDiffusion supports consistent editing of both generated and real images and facilitates user interaction in both conditional and unconditional settings, as shown in Figure 1. Moreover, while sampling efficiency was not our primary objective, our approach yields faster sampling and training convergence due to the shorter trajectories between stages (Figure 2), resulting in fewer denoising steps and less computational cost (Figure 1). For instance, LDM is trained using 1k steps; in contrast, ToddlerDiffusion could be trained using only 10 steps with minimal impact on generation fidelity. To the best of our knowledge, ToddlerDiffusion is the first work to successfully integrate and achieve advancements across multiple domains, including controllability, consistent editing, sampling and training efficiency, and interactivity, within a unified framework, as shown in Figure 1. Our extensive experiments on datasets such as LSUN-Churches (Yu et al., 2015), ImageNet, and CelebA-HQ (Karras et al., 2017) demonstrate the effectiveness of this approach, consistently outperforming existing methods. ToddlerDiffusion not only redefines the image generation process but also paves the way for future research into more structured, interpretable, and efficient generative models.

Our contributions can be succinctly summarized as follows:

- We introduce a cascaded structured image generation pipeline, denoted as ToddlerDiffusion, that systematically generates a chain of interpretable stages leading to the final image.
- Leveraging the Schrödinger Bridge enables us to train the model from scratch using a minimal number of steps: 10 steps only, in addition to, performing on the bar compared to LDM while being $2\times$ faster, using $3\times$ smaller architecture.
- ToddlerDiffusion provides robust editing and interaction capabilities for free; without requiring any manual annotations.
- Introduce a novel formulation for sketch generation that achieves better performance than LDM while using $41\times$ smaller architecture.

## 2 TODDLERDIFFUSION

We introduce ToddlerDiffusion, which strategically decomposes the RGB image generation task into more straightforward and manageable subtasks, such as sketch/edge and palette modalities. In Section 2.1, we detail the formulation of each stage. By breaking down the generation process into simpler components, our model not only allows for a more compact architecture but also enables dynamic user interaction with the system, as illustrated in Figure 1. Notably, these advancements are achieved without relying on manual ground truth annotations, instead utilizing human-free guidance for enhanced efficiency and practical applicability (Section 2.2). Lastly, we demonstrate how our method facilitates robust, consistent edits across stages (Section 2.4).

### 2.1 TODDLER GROWTH

The development of our method is inspired by child's growth, where the learning process in their brains is decomposed into stages and developed step-by-step (Stevens, 2012; Marr, 2010; 1974). By analogy, we decompose the 2D image synthesis into different, more straightforward stages. We employ the sketch and palette as intermediate stages, resulting three cascaded stages approach. However, our approach is generic and seamlessly we can add or remove intermediate steps, based

on the application needs. Mainly, the pipeline can be clustered into two main modules: 1) The unconditional module, that generates the first modality, e.g., sketches, based on pure noise. 2) The conditional module, that condition on the previous modality to generate the next intermediate modality, e.g., palette, or the final RGB image. Both modules can be implemented using Latent Diffusion Models (LDM), where unconditional LDM will start from noise to generate sketches, then by employing the concatenation conditioning mechanism we can condition on our spatial intermediate modalities; sketches and palettes. We instead adopt a more principled approach by employing the Schrödinger Bridge to determine the optimal path between modalities at each stage. Unlike simple concatenation, which starts each stage from a state of pure noise, the Schrödinger Bridge enables us to begin each stage from a meaningful signal with a higher Signal-to-Noise Ratio (SNR).

**Schrödinger Bridge.** The Schrödinger Bridge problem arises from an intriguing connection between statistical physics and probability theory. At its core, it addresses the task of finding the most likely stochastic process that evolves between two probability distributions over time, $\rho_0$ and $\rho_T$, governed by Brownian motion, while minimizing the control cost. Mathematically, it could be written as: $P^* = \underset{P \in \mathcal{P}}{\arg\min} \, \mathrm{KL}(P \,\|\, W)$, where $P$ is the probability path that minimizes the Kullback-Leibler (KL) divergence with respect to a reference Brownian motion $W$, and subject to marginal constrains; $P_0 = \rho_0$ and $P_T = \rho_T$. The process can be described as a stochastic differential equation (SDE):

$$dX_t = v(t, X_t)\, dt + \sigma \, dW_t, \tag{1}$$

where $X_t$ represents the state at time $t$, $v(t, X_t)$ is the optimal drift that minimizes the KL divergence and $dW_t$ is the Wiener process (Brownian motion).

**Conditional Module.** By analogy, the two probability distributions $\rho_T$ and $\rho_0$ represent the distributions of features from simpler (sketch) and more complex (RGB) modalities, respectively. In this context, the goal of the Schrödinger Bridge is to find the optimal stochastic process that transforms the simpler modality (e.g., sketch) into the more complex one (e.g., RGB image) while minimizing a divergence measure like KL divergence while respecting the boundaries. Given a condition $y \in \mathbb{R}^{H \times W \times 3}$, that could be sketch, palette, or any other modality, we define the forward process as an image-to-image translation and formulate the problem using the discrete version of schrödinger bridge as follows:

$$x_t = (1 - \alpha_t)x_0^i + \alpha_t y^i + \sigma_t^2 \epsilon_t, \qquad : \sigma_t^2 = \alpha_t - \alpha_t^2, \tag{2}$$

where $i$ is the stage number, $x_0$ and $y$ are the boundaries of the bridge, $\sigma_t^2$ controls the noise term, and $\alpha_t$ blends the two domains. In the SDE, the term "$v(t, X_t)$" is the deterministic drift, while "$dW_t$" represents the diffusion term. In contrast, we follow the DDPM formulation, which works in the discrete space. Therefore, by analogy, the terms "$\alpha_t x_0 + (1 - \alpha_t)y$" and "$\sigma_t^2 \epsilon_t$" represent the drift and diffusion terms, respectively. $\sigma_t^2$ represents uncertainties, following a sphere formulation that constructs a bridge between the two domains. Whereas, at the edge of the bridge, we are fully confident we are in one of the distributions $\rho_T$ and $\rho_0$, thus $\sigma_t^2 = 0$. In contrast, the uncertainties increase while moving away from the edges, reaching their maximum value in the middle between the two domains, forming the bridge shape. Eq. 2 forms the skeleton for $2^{nd}$ and $3^{rd}$ stages, where:

$$x_0^i = \begin{cases} x_0^p = \mathcal{F}_p(x_0^{rgb}) & : i = 2 \quad \text{(Palette)} \\ x_0^{rgb} & : i = 3 \quad \text{(Detailed Image)} \end{cases} \qquad y^i = \begin{cases} x_0^s = \mathcal{F}_s(x_0^{rgb}) & : i = 2 \quad \text{(Palette)} \\ x_0^p = \mathcal{F}_p(x_0^{rgb}) & : i = 3 \quad \text{(Detailed Image)} \end{cases} \tag{3}$$

where $i$ is the stage number, $\mathcal{F}_p$ and $\mathcal{F}_s$ are palette and sketch functions that generate the pseudo-GT, respectively, detailed in Section 2.2.

**Unconditional Module: $1^{st}$ Stage (Sketch).** This stage aims to generate abstract contours $x_0^s \in \mathbb{R}^{H \times W}$ starting from just pure noise or label/text conditions. In both cases, we name this module unconditional as it does not condition on a previous modality because it will generate the first modality, the sketch. One possible solution is to utilize the original LDM, which is not aligned with the sketch nature, as shown in Figure 3, case A. More specifically, we are starting from a pure noise distribution $\mathbf{x}_T$ and want to learn the inverse mapping $q(\mathbf{x}_{t-1} \mid \mathbf{x}_t)$ to be able to sample real data $x_0^s$. The issue relies upon the vast gap

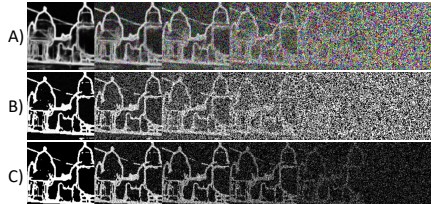

Figure 3: Comparison between different formulations for the $1^{st}$ stage. This depicts the forward process for each formulation.

between the two domains: $x_T$ and $x_0$. Following (Kingma et al., 2021), this can be interpreted as

signal-to-noise ratio (SNR), where $\text{SNR}(t) = \frac{\alpha_t}{\sigma_t^2}$. Consequently, at the beginning $(t = T)$, $\alpha_T$ yields to 0, therefore $\text{SNR}(T)=0$. On the contrary, to fill this gap, we propose a unique formulation tailored for sketch generation that adapts Eq. 2. Specifically, we introduced two changes to Eq. 2. First, we replaced the Gaussian distribution $\epsilon_t$ with a discretized version, the Bernoulli distribution, as shown in case B, Figure 3. However, this is not optimized enough for the sparsity nature of the sketch, where more than 80% of the sketch is black pixels. Thus, we set our initial condition to $y = \mathcal{F}_d(x_0^s)$, where $\mathcal{F}_d$ is a dropout function that takes the GT sketch $x_0^s$ and generates more sparse version. To align our design with the sketch nature, as shown in Figure 3, part C, we changed $\sigma_t^2$ to follow a linear trend instead of the bridge shape by setting the variance peak at $x_T$. We have to make sure to set the variance peak to a small number due to the sparsity of the sketch. This could be interpreted as adding brighter points on a black canvas to act as control points during the progressive steps $(T \to 0)$ while converging to form the contours, as shown in Figure 3, part C. More details about the sparsity level importance are shown in Section A.6.

## 2.2 Toddler Guidance

A crucial factor for the success of our framework, ToddlerDiffusion, is obtaining accurate and human-free guidance for the intermediate stages. Two modules, $\mathcal{F}_s$ and $\mathcal{F}_p$, are employed to generate contours/sketches and palettes, respectively.

**Sketch.** The first module, $x_0^s = \mathcal{F}_s(x_0^{rgb})$, serves as a sketch or edge predictor, where $x_0^{rgb} \in \mathbb{R}^{H \times W \times 3}$ is the ground-truth RGB image, and $x_0^s \in \mathbb{R}^{H \times W}$ is the generated sketch. For instance, $\mathcal{F}_s$ could be PidiNet (Su et al., 2021) or Edter (Pu et al., 2022) for sketch generation, or Canny (Canny, 1986) and Laplacian (Wang, 2007) edge detectors.

**Palette.** The palette can be obtained without human intervention by pixelating the ground-truth RGB image $x_0^p = \mathcal{F}_p(x_0^{rgb})$, where $x_0^p \in \mathbb{R}^{H \times W \times 3}$ is the palette image, and $\mathcal{F}_p$ will be the pixelation function. However, we introduce a more realistic way to have high-quality palettes for free by first generating the segments from the RGB image using FastSAM (Zhao et al., 2023b). Then, a simple color detector module is utilized to get the dominant color, e.g., the median color per segment. This design choice is further analyzed in Section A.6. More details are mentioned in the Appendix.

## 2.3 Training Objective & Reverse Process

**Training Formulation.** We adapt the conventional diffusion models' learning objective (Ho et al., 2020; Sohl-Dickstein et al., 2015), i.e., Variational Lower Bound (ELBO), to include our new condition $y$, whereas, each marginal distribution has to be conditioned on $y$, as follows:

$$\mathcal{L}_{ELBO} = -\mathbb{E}_q \big( D_{KL}(q(x_T|x_0^i, y^i)||p(x_T|y^i)) + \sum_{t=2}^{T} D_{KL}(q(x_{t-1}|x_t, x_0^i, y^i)||p_\theta(x_{t-1}|x_t, y^i)) - \log p_\theta(x_0^i|x_1, y^i) \big),$$

(4)

where $p_\theta$ is a function approximator intended to predict from $x_t$. Using Bayes'rule, we can formulate the reverse process as follows:

$$q\left(x_{t-1} \mid x_t, x_0^i, y^i\right) = \mathcal{N}\left(x_{t-1}; \tilde{\mu}\left(x_t, x_0^i, y^i\right), \tilde{\sigma}_t^2 \mathbf{I}\right)$$

(5)

$$\tilde{\mu}_t(x_t, x_0^i, y^i) = \frac{\sigma_{t-1}^2}{\sigma_t^2} \frac{1 - \alpha_t}{1 - \alpha_{t-1}} x_t + (1 - \alpha_{t-1}(1 - \frac{\sigma_{t-1}^2 (1 - \alpha_t)^2}{\sigma_t^2 (1 - \alpha_{t-1})^2})) x_0^i + (\alpha_{t-1} - \alpha_t \frac{1 - \alpha_t}{1 - \alpha_{t-1}} \frac{\sigma_{t-1}^2}{\sigma_t^2}) y^i$$

(6)

$$\tilde{\sigma}_t^2 = \sigma_{t-1}^2 - \frac{\sigma_{t-1}^4}{\sigma_t^2} \frac{(1 - \alpha_t)^2}{(1 - \alpha_{t-1})^2}$$

(7)

The derivation for Eq. 6 and Eq. 7 is detailed in Appendix A.1. The overall training and sampling pipelines are summarized in Appendix A.5 Algorithms 1 and 2, respectively.

**Training Objectives.** Previous work (Rombach et al., 2022) showed that learning directly the noise $\epsilon$ enhances the performance compared to predicting the $x_0$ directly. This is intuitive as predicting the difference $\epsilon_t \propto x_t - x_0$ is easier than predicting the original complicated $x_0$. However, in our case, this is not valid due to the conditioned domain $y$, which makes predicting the difference much harder, as shown in Eq. 8

$$x_t - x_0 = \alpha_t(y - x_0) + \sigma_t \epsilon_t$$

(8)

Accordingly, we directly predict the $x_0$. Please refer to Section A.6 for a detailed analysis.

### 2.4 TODDLER CONSISTENCY

Our novel framework, ToddlerDiffusion, inherently provides interactivity, allowing users to observe and engage with the intermediate outputs of various modalities. This interactivity is further strengthened by ensuring consistency, a crucial element for effective control and editing capabilities.

**Generated Images Editing.** To ensure consistency during editing, we store the noise $\epsilon_t$ and the intermediate modalities used in generating the initial reference image, creating a memory that will help us achieve outstanding consistency. Throughout the editing session, the memory, that was constructed while generating the reference image, remains fixed until the user resets it by generating a new reference image. Figure 10 demonstrates our framework's editing capabilities, highlighting the strong consistency it achieves. Additional editing examples can be found in our demo.

**Real Images Editing.** Real image editing is much more challenging as we cannot access intermediate modalities and noise. Therefore, given a real image $x_0^{rgb}$, first, we run $\mathcal{F}_s(x_0^{rgb})$ and $\mathcal{F}_p(x_0^{rgb})$ to get the corresponding sketch $x_0^s$ and palette $x_0^p$, respectively. Then, we utilize the recent DDPM inversion technique (Huberman-Spiegelglas et al., 2024) to store the noise $\epsilon_t$. More specifically, to get the noise, we have first to run our forward process, Eq. 2, using random noise $\epsilon_t$ and store the corresponding intermediate states $x_t$. Then, run our reverse process, Eq. 6, to try to recover the original image. During the reverse process, the model will predict $\tilde{\mu}_t(x_t, x_0^i, y^i)$ using Eq. 6. Then, our goal is that given the predicted $\tilde{\mu}_t$, the predicted conditions $y$ and the stored intermediate states $x_{t-1}$, we want to estimate the noise $\epsilon_t$ Following these two steps, we construct our memory, which will be used similarly to edit real images.

## 3 EXPERIMENTAL RESULTS

To probe the effectiveness of our proposed framework, we first compare our approach against LDM when trained from scratch on unconditional datasets, LSUN-Churches and CelebHQ, and conditional ones, ImageNet. Then, we show that our approach can leverage from the high-quality SD-v1.5 weights, enabling the model to be conditional on new modalities seamlessly without adding additional modules (Zhang et al., 2023; Zhao et al., 2024) or adapters (Mou et al., 2024).

### 3.1 TODDLERDIFFUSION FROM SCRATCH

In this section, first we will holistically compare our method, ToddlerDiffusion, against LDM regarding generation quality and efficiency, e.g., convergence rate and number of denoising steps. Then, we will compare our conditioning ability with ControlNet and SDEdit. Generally, ToddlerDiffusion consists of three cascaded stages: sketch, palette, and RGB generation. However, in the following section, we will show the versatility of our method by omitting the $1^{st}$ or the $2^{nd}$ stages, enabling our method to start from a reference sketch or palette, respectively, instead of generated ones.

**Unconditional Generation.** We evaluated our framework, ToddlerDiffusion, against LDM on two unconditional datasets, i.e., LSUN-Churches (Yu et al., 2015) and Celeb datasets (Karras et al., 2017). For a fair comparison, and as we notice challenges in replicating LDM results[1], we have reproduced LDM results. As shown in Table 1, we boost the LDM performance by 1 FID point on the two datasets, using 3x smaller architecture and 2x faster sampling, as shown in Table 2.

**Efficient Architecture.** While the primary focus of this paper is not on designing an efficient architecture, our method inherently benefits from architectural efficiency due to the shorter trajectory between stages, as illustrated in Figure 2. The key insight is that ToddlerDiffusion simplifies each generation task by breaking it into modality-specific stages (e.g., from sketch to palette). This approach allows us to leverage significantly smaller networks at each stage because solving the problem is easier, where the distributions of the domains, such as sketches and palettes, are much closer than the distribution between pure noise and RGB images. As a result, the neural network at each stage can be more lightweight. Furthermore, as shown in Table 2, we operate directly in the image space for the first two stages (i.e., noise to sketch and sketch to palette). This eliminates the need for complex encoder-decoder architectures, such as VQGAN, typically used to generate

---

[1]See issues 325, 262, 90, 142, 30, and 138 in the LDM GitHub repository.

Table 1: Benchmark results on three datasets; CelebHQ Karras et al. (2017), LSUN-Churches Yu et al. (2015) and ImageNet Deng et al. (2009).

| Dataset | Method | Epochs | FID ↓ | KID ↓ | Prec. ↑ | Recall ↑ |
|---------|--------|--------|-------|-------|---------|----------|
| **CelebHQ** | LDM | 600 | 8.15 | 0.013 | 0.52 | 0.41 |
| | Ours | 600 | **7.10** | **0.009** | **0.61** | **0.47** |
| **Churches** | LDM | 250 | 7.30 | 0.009 | 0.59 | 0.39 |
| | Ours | 250 | **6.19** | **0.005** | **0.71** | **0.44** |
| **ImageNet** | LDM | 350 | 8.55 | 0.015 | 0.51 | 0.32 |
| | Ours | 350 | **7.8** | **0.010** | **0.58** | **0.40** |

Table 2: A detailed analysis of our method complexities compared to LDM on the CelebHQ dataset Karras et al. (2017). We report the training and sampling throughput per each stage. For efficiency, the first two stages are operating on the pixel space. Therefore, the VQGAN encoder and decoder are not used (N/A). The training time is reported till convergence, i.e, 600 epochs, using 4 A100.

| Stage | VQGAN # Param | UNet # Param | Train Time | Samp. Time |
|-------|---------------|--------------|------------|------------|
| 1) Noise → Sketch | N/A | 1.9M | 1 hr | 85 fps |
| 2) Sketch → Palette | N/A | 1.9M | 1 hr | 85 fps |
| 3) Palette → RGB | 55.3M | 101M | 24 hr | 2.3 fps |
| Ours (3 Stages) | 55.3M | **104.8M** | **26hr** | **2.27fps** |
| LDM | 55.3M | 263M | 38 hr | 1.21 fps |

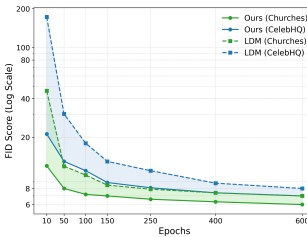

Figure 4: Training convergence comparison between ToddlerDiffusion and LDM.

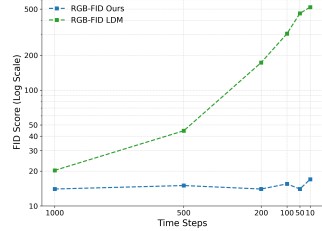

Figure 5: Trimming denoising steps ablation study during training and sampling.

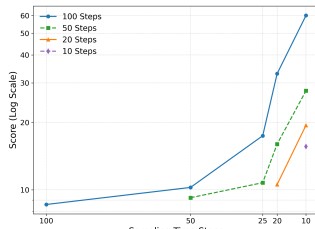

Figure 6: Training steps ablation study on LSUN-Churches dataset.

detailed images from latent representations. Combining these factors, ToddlerDiffusion achieves a 3x reduction in network size despite cascading across three stages.

**Faster Convergence.** In addition to outperforming LDM across multiple datasets and metrics, as shown in Table 1, ToddlerDiffusion exhibits a significantly faster convergence rate. We analyzed the convergence behavior by reporting the FID score every 50 epochs, as shown in Figure 4. Remarkably, our method achieves the same performance after just 50 epochs as LDM does after 150 epochs. Furthermore, after 150 epochs, ToddlerDiffusion matches the performance LDM achieves after 500 epochs. This substantial improvement in convergence speed highlights the strength of our approach in efficiently navigating the feature space between stages. It's important to note that faster convergence was not an explicit target of our method but rather an emergent property of the pipeline's design. Other techniques tailored explicitly for improving convergence speed, e.g., (Hang et al., 2023), can be considered orthogonal and complementary to our method.

**Trimming Denoising Steps.** Another unique characteristic of ToddlerDiffusion is its ability to reduce the number of denoising steps during both training and sampling without significantly impacting performance. As shown in Figure 5, our framework maintains consistent and robust results when the number of steps is reduced from 1000 to 100. In contrast, the performance of LDM degrades drastically under the same conditions. ToddlerDiffusion can trim denoising steps while maintaining a good Signal-to-Noise Ratio (SNR), particularly for larger time steps, as further discussed in Section 2.1. Additionally, Figure 6 highlights another intriguing capability of our method: models trained with fewer denoising steps outperform those trained with larger step counts when later using techniques like DDIM (Song et al., 2020a) to reduce sampling steps. For example, as seen in Figure 6, the blue curve (trained with 100 steps) achieves an FID score of around 60 when using only ten steps for sampling. However, if we initially train the model with just ten steps, as represented by the purple curve, the performance improves significantly, achieving an FID score of 15. This demonstrates the efficiency and adaptability of ToddlerDiffusion, even when minimizing the denoising process.

**Class-Label Conditional Generation.** To holistically evaluate the performance of ToddlerDiffusion, we tested it on the widely-used ImageNet dataset, where the condition type is a class label. Due to limited resources, we conducted our experiments on ImageNet-100, a subset of ImageNet containing 100 classes instead of the full 1,000. As shown in Table 1, ToddlerDiffusion performs better than LDM on key metrics such as FID, KID, Precision, and Recall.

**Controllability.** We evaluated our editing capabilities against two notable methods: SDEdit (Meng et al., 2021) and ControlNet (Zhang et al., 2023). As illustrated in Figure 7, SDEdit struggles when conditioned on binary sketches, which is expected given that it is a training-free approach designed to operate with conditions closer to the target domain (i.e., RGB images). Consequently, compared to ControlNet, a training-based method that adds copy of the encoder network, our approach

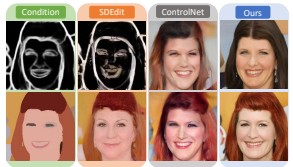
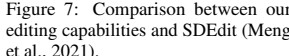
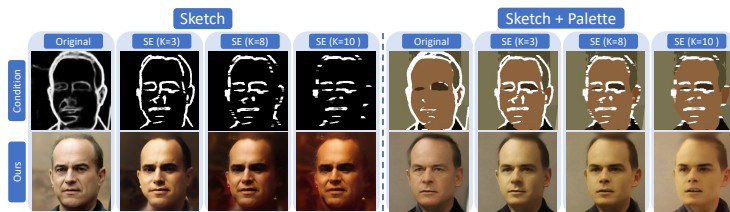

Figure 7: Comparison between our editing capabilities and SDEdit (Meng et al., 2021).

Figure 8: Sketch robustness analysis.

demonstrates superior faithfulness to the sketch condition, consistency across the generated images, and overall realism. Starting from the converged model train on the CelebHQ dataset for 600 epochs, we train our method and ControlNet for and additional 50 epochs with the sketch as a condition. Despite ControlNet's larger model size, our method achieves these improvements without adding additional parameters. Quantitatively, we achieved an FID score of 7, outperforming SDEdit and ControlNet, which achieved 15 and 9, respectively.

## 3.2 ROBUSTNESS ANALYSIS

**Label Robustness.** Beyond these quality comparisons, we also conducted a robustness analysis to assess our model's behavior when faced with inconsistencies between conditions. Specifically, we deliberately provided incorrect class labels for the same sketch.

This mismatched the sketch (geometry) and the desired class label (style). Despite the inconsistent conditions, as illustrated by the off-diagonal images in Figure 9, our model consistently adheres to the geometry defined by the sketch while attempting to match the style of the misassigned category, demonstrating the robustness of our approach. Successful cases are highlighted in yellow, and even in failure cases (highlighted in red), our model avoids catastrophic errors, never hallucinating by completely ignoring either the geometry or the style. More examples can be seen in Appendix B.3.

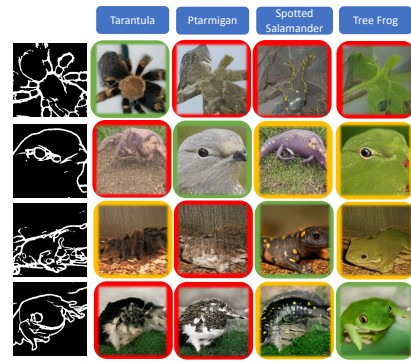

**Sketch Robustness.** To further validate the robustness of ToddlerDiffusion, we designed an aggressive robustness test aimed at evaluating the model's ability to handle sketch perturbations. Specifically, we generated three variants of the input sketches using the Structured Edge (SE) detector (Dollár & Zitnick, 2014), varying the kernel size $K$ to introduce different detail and noise levels. As shown in Figure 8, our model consistently

Figure 9: Our results on the ImageNet dataset. The first column depicts the input sketch. The diagonal (green) represents the generated output that follows the sketch and the GT label. The off-diagonal images show the output given the sketch and inconsistent label. These adversarial labels show our model's robustness, where the successful cases are in yellow, and the failure ones are in red.

demonstrates strong robustness despite these perturbations being absent during training. The left part of Figure 8 depicts our model's output using the $2^{nd}$ stage only (Sketch to RGB), while the right part demonstrates the output when using both the $2^{nd}$ and $3^{rd}$ stages. Even with significant variations in the sketches, ToddlerDiffusion maintains high fidelity to both the geometry and style, underscoring its ability to adapt to unseen input distortions without compromising output quality.

## 3.3 EDITING CAPABILITIES

**Generated Images Editing.** In Figure 10, we showcase the editing capabilities starting from a generated sketch and RGB image (A). Our method allows for the removal of artifacts or unwanted elements, highlighted in red (B), the addition of new content, shown in yellow (C-D), and modifications to existing content, marked in green (E-F), simply by manipulating the underlying sketch. These consistent edits are supported in unconditional and conditional generation scenarios. While sketch-based conditional generation naturally lends itself to such control, ToddlerDiffusion extends this capability to unconditional settings and other conditional models like Text-to-Image. In these cases, we refer to this property as "interactivity," allowing users to seamlessly edit through intermediate modalities (sketch and palette) regardless of the original input format, whether class-label or text.

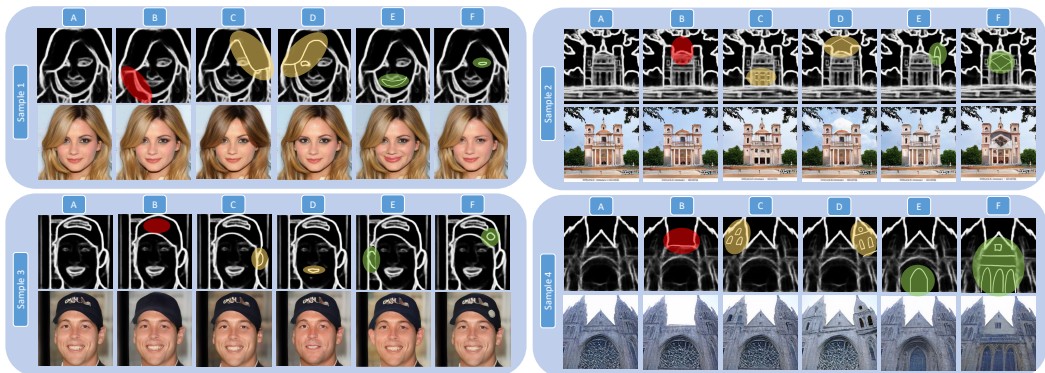

Figure 10: Controllability ability of our framework, ToddlerDiffusion. Starting from the generated sketch and RGB image (A), we can remove artifacts or undesired parts in red (B), add new content in yellow (C-D), and edit the existing content in green (E-F) by manipulating the sketch.

**Real Images Editing.** As explained in Section 2.4, we ensure consistent edits for real images by storing the noise used during the generation process for later edits. The primary challenge with real image editing is that we don't have access to the noise values at each denoising step, as our model didn't generate the image. To address this, we employ DDPM-inversion (Huberman-Spiegelglas et al., 2024) to reconstruct the intermediate noise. As illustrated in Figure 1, our model achieves consistent edits on real reference images. In Appendix B.2, we compare against SEED-X (Ge et al., 2024), and our model achieves more consistent edits, as shown.

## 3.4 INCORPORATING TODDLERDIFFUSION INTO STABLE DIFFUSION

In this section, we will answer an important question: "Can our method be integrated seamlessly into the existing powerful Stable-Diffusion architecture?" This question is crucial as follows: 1) Despite our approach's great capabilities and performance, leveraging the available powerful weights trained on large-scale datasets such as SD is still important. 2) Our approach follows different forward and reverse processes, Eq. 2 and Eq. 6, respectively, which could make it challenging to leverage weights that are trained using different training objectives.

**Integration Recipe.** This section will detail the recipe for successfully integrating SD weights into our approach. we initially adopted $x_0$ as the training objective instead of the original SD objective, $\epsilon$, and employed Low-Rank Adaptation (LoRA) for fine-tuning. However, we observed that this approach does not converge easily due to the significant disparity between the pre-trained SD model's training objective and the new objective. Consequently, we opted to fine-tune the entire model. Remarkably, after just 10k iterations, the model generated meaningful images that aligned perfectly with the input conditions, such as sketches or palettes. More qualitative results can be seen in Figure 24 in Appendix B.6

**Controllability.** We convert our method to a "sketch2RGB" by omitting the $1^{st}$ stage to show our controllability capabilities. By doing this, we can compare our model against the vast tailored sketch-based models built on top of SD-v1.5. However, these methods have been trained for a long time on large-scale datasets. Thus, for a fair comparison, we retrain them using CelebHQ datasets for 5 epochs starting from SD-v1.5 weights. As demonstrated in Appendix B.5, despite this is not our main focus, our method outperforms conditioning conditioning methods without adding additional modules (Zhang et al., 2023; Zhao et al., 2024) or adapters (Mou et al., 2024).

## 4 DISCUSSION AND FUTURE WORK

In this section, we will discuss some possible future work that shows the versatility of ToddlerDiffusion and its potential to drive innovation across image, video, and 3D tasks. Detailed future work is discussed in Appendix A.9.

**Distill the knowledge from substage features for perception tasks.** The intermediate outputs from Toddler Diffusion (e.g., the sketch stage) can be reused as strong geometric priors for perception tasks by distilling the knowledge into perception networks to enforce geometric or perceptual constraints, making them structure-aware and geometry-guided aligned (Zhao et al., 2023a).

**Reusing Diffusion Features as High- and Low-Frequency Representations.** Toddler Diffusion's staged outputs inherently capture distinct frequency information, with the sketch stage emphasizing low-frequency global structures and geometry and the palette and RGB stages capturing high-frequency details like textures. These features can be repurposed into a dual-branch network (Bi et al.; Wei et al., 2024): one branch focusing on low-frequency representations for tasks requiring global context (e.g., semantic segmentation, object localization) and the other specializing in high-frequency details for tasks like texture classification, instance segmentation, or boundary refinement. Integrating this knowledge into VLMs can improve performance and generalization for perception tasks.

**Addressing Limitations in T2I Models.** A significant challenge for text-to-image (T2I) models lies in handling complex prompts involving counting or spatial relations (Bakr et al., 2023; Ghosh et al., 2024). Fixing these issues in the sketch or abstract stage will be easier and simpler due to its lightweight. In addition, the subsequent stages (e.g., sketch-to-RGB) will remain unaffected and reusable. This approach allows targeted improvements without retraining/finetuning the entire pipeline, significantly reducing computational costs.

**Modular Prompt Handling for T2I Models.** Toddler Diffusion can simplify complex text prompts by decomposing them into modular components mapped to specific stages. For example, a detailed prompt can be split into geometry-focused instructions for the sketch stage and color-specific details for the sketch-to-palette stage. In addition, we can add more stages and specific prompts such as depth, e.g., $obj_1$ closer than $obj_2$. This enables a more focused and interpretable generation, particularly for long, detailed prompts with intricate requirements.

For instance: Given a complex prompt: "A bustling city street during sunset with a red double-decker bus on the right, a blue car parked on the left, people walking on the sidewalk, and sunlight casting long shadows on the ground." will be split into:

- Structure-Focused prompt: "A city street with a double-decker bus on the right, a car parked on the left, and people walking on the sidewalk."
- Palette-Focused prompt: "The double-decker bus is red, the car is blue, the sky shows a sunset with shadows on the ground."

**Video Generation in Abstract Domains.** Generating videos directly in the RGB domain is computationally expensive and prone to inconsistencies. Toddler Diffusion can split the problem into geometry consistency and stylistic consistency. Specifically, it first generates videos in an abstract domain (e.g., sketches), ensuring geometric consistency across frames. Then, these sketches can be refined into RGB frames using another stage, Sketch-to-RGB, conditioned on a reference frame for color and style consistency. This two-step process simplifies the complex task of maintaining temporal and stylistic coherence in video generation.

## 5 CONCLUSION

ToddlerDiffusion introduces a paradigm shift in image generation by breaking down the complex task into more manageable, modality-specific stages. Inspired by the natural learning process of a child, our framework elegantly mirrors how human artists progress from rough sketches to fully detailed images. This structured approach allows us to simplify each stage of the generation process, enabling the use of smaller architectures and achieving efficient, high-quality results. Despite the potential challenges introduced by multi-stage frameworks, such as increased parameter counts and error accumulation, ToddlerDiffusion overcomes these obstacles through the strategic use of the Schrödinger Bridge, which ensures smooth transitions between modalities. By maintaining a high Signal-to-Noise Ratio (SNR) at each stage, we enable stable, effective progression, minimizing the number of denoising steps required during both training and sampling. This results in faster convergence, lower computational cost, and enhanced sampling efficiency. Moreover, ToddlerDiffusion provides a level of control and interaction that is unprecedented in current diffusion-based models, offering consistent editing capabilities across both generated and real images. Through extensive experiments on multiple datasets, we have shown that ToddlerDiffusion outperforms existing state-of-the-art models like LDM in terms of quality, interactivity, and speed. The flexibility and versatility of our approach make it applicable to a wide range of tasks, and its emergent properties, such as faster convergence and improved editing control, further solidify its potential for real-world applications.

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

# A  TODDLERDIFFUSION METHOD (MORE DETAILS)

## A.1  DERIVATIONS

From the forward process, we have:

$$x_t = (1 - \alpha_t)x_0 + \alpha_t y + \sigma_t \epsilon_t \tag{9}$$

$$x_{t-1} = (1 - \alpha_{t-1})x_0 + \alpha_{t-1} y + \sigma_{t-1} \epsilon_{t-1} \tag{10}$$

From equation 10, we get:

$$x_0 = \frac{1}{1 - \alpha_{t-1}} x_{t-1} - \frac{\alpha_{t-1}}{1 - \alpha_{t-1}} y - \frac{\sigma_{t-1}}{1 - \alpha_{t-1}} \epsilon_{t-1} \tag{11}$$

Inserting equation 11 into equation 9, we get:

$$q(x_t \mid x_{t-1}, x_0) = \frac{1 - \alpha_t}{1 - \alpha_{t-1}} x_{t-1} - \frac{(1 - \alpha_t)\alpha_{t-1}}{1 - \alpha_{t-1}} y$$
$$+ \alpha_t y - \frac{(1 - \alpha_t)\sigma_{t-1}}{1 - \alpha_{t-1}} \epsilon_{t-1} + \sigma_t \epsilon_t \tag{12}$$

The reverse process follows:

$$q\left(x_{t-1} \mid x_t, x_0, y\right) = \mathcal{N}\left(x_{t-1}; \tilde{\mu}\left(x_t, x_0, y\right), \tilde{\sigma}_t^2 \mathbf{I}\right) \tag{13}$$

Using Bayes' rule:

$$q\left(x_{t-1} \mid x_t, x_0, y\right) = q(x_t \mid x_{t-1}, x_0) \frac{q(x_{t-1} \mid x_0)}{q(x_t \mid x_0)} \tag{14}$$

Plugging equation 9, equation 10, and equation 12 into equation 14, yields:

$$\tilde{\mu}_t(x_t, x_0, y) = \frac{\sigma_{t-1}^2}{\sigma_t^2} \frac{1 - \alpha_t}{1 - \alpha_{t-1}} x_t + (1 - \alpha_{t-1}\left(1 - \frac{\sigma_{t-1}^2}{\sigma_t^2} \frac{(1 - \alpha_t)^2}{(1 - \alpha_{t-1})^2}\right))x_0$$
$$+ (\alpha_{t-1} - \alpha_t \frac{(1 - \alpha_t)}{(1 - \alpha_{t-1})} \frac{\sigma_{t-1}^2}{\sigma_t^2})y \tag{15}$$

$$\tilde{\sigma}_t^2 = \sigma_{t-1}^2 - \frac{\sigma_{t-1}^4}{\sigma_t^2} \frac{(1 - \alpha_t)^2}{(1 - \alpha_{t-1})^2} \tag{16}$$

Finally, we can apply the DDIM non-Markovian forward processes.
In return, only the $x_0$ coefficient will be affected.

## A.2  ARCHITECTURE DETAILS

Figure 11 depicts the details of our architecture. We adopt the LDM architecture, which consists of two main blocks: 1) VQGAN Encoder-Decoder to convert the image into low-cost latent space. 2) Unet that responsible for predicting $x_{t-1}$. Figure 11 shows that the condition $y$ is encoded into $x_t$. However, it could be optionally concatenated to $x_t$. In addition, shown in orange and green are the $\sigma^2$ and $\alpha$, respectively. Consequently, Figure 12 depicts an abstract flow for the sampling mode, where we show how our approach works starting from random noise till generating the final RGB image.

## A.3  $1^{st}$ STAGE: ABSTRACT STRUCTURE

**Sketch FID.** A discrepancy exists between the reported conventional FID (RGB-FID), trained on ImageNet (Deng et al., 2009), and qualitative results, as illustrated in Figure 13. This discrepancy (Ge et al., 2020) may arise from differences between the training data (RGB images) and the evaluation

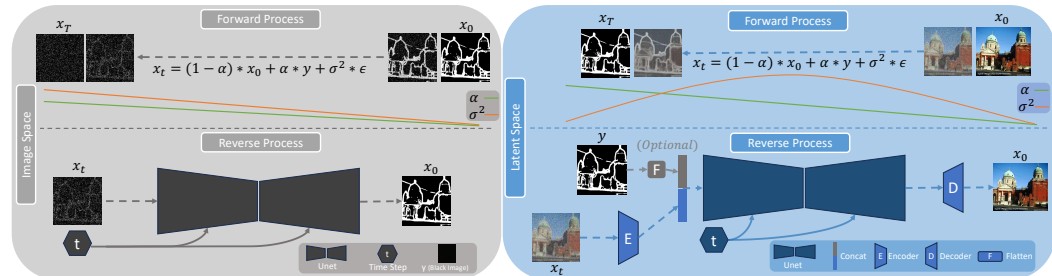

Figure 11: An overview of proposed architecture, dubbed ToddlerDiffusion. The left block demonstrates the first stage, which generates a sketch unconditionally. Due to our efficient formulation, this stage operates in the image space on $64 \times 64$ resolution. The right module depicts the third stage, which generates an RGB image given a sketch only or both sketch and palette.

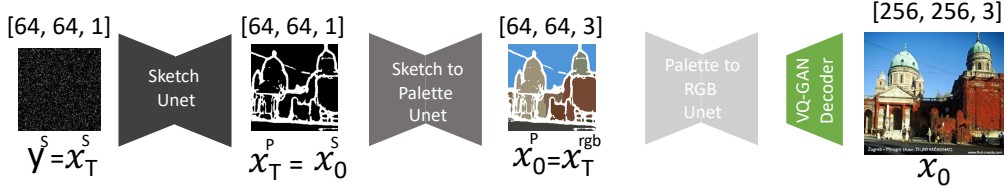

Figure 12: An abstract flow for the sampling mode, where we show how our approach works starting from random noise till generating the final RGB image.

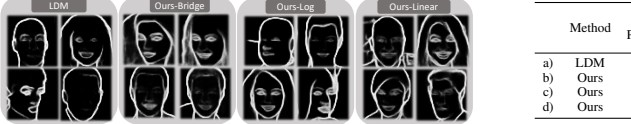

| | Method | Unet Parameters | Training Time (Min:Sec)↓ | Sampling Time (FPS)↑ | Noise Scheduler | Sketch FID ↓ | RGB FID ↓ |
|---|---|---|---|---|---|---|---|
| a) | LDM | 187 M | 4:05 | 2.2 | Linear | 126.4 | 23.5 |
| b) | Ours | 1.9 M | 0:18 | 53 | Bridge | 110.5 | 35.2 |
| c) | Ours | 1.9 M | 0:18 | 53 | Log | 61.2 | 15.6 |
| d) | Ours | **1.9 M** | **0:18** | **53** | Linear | **58.8** | **15.1** |

Figure 13: Benchmarking results for the $1^{st} stage$ on the CelebA-HQ dataset (Karras et al., 2017). The training time is calculated per epoch using 4 NVIDIA RTX A6000. The sampling time is calculated per frame using one NVIDIA RTX A6000 with a batch size equals 32.

data (binary images). To bridge this gap, we introduce Sketch-FID by re-training the inception model (Szegedy et al., 2015) on a sketch version of the ImageNet dataset. We generate sketches for ImageNet RGB images using PidiNet (Su et al., 2021) and train the inception model on the conventional classification task.

**Noise Scheduler.** In the $1^{st}$ stage, where our starting point $y$ is a black image (Section 2.1), designing an appropriate noise scheduler is crucial. The bridge noise scheduler is intuitively unsuitable, as it eliminates randomness by adding no noise at both edges, fixing the starting point to a black image. This hypothesis is supported by empirical results in Figure 13, row b, where the model outputs random patterns. We explored linear and logarithmic schedulers, finding the linear schedule superior, yielding Sketch-FID scores of 15.19 and 18.47, respectively (Figure 13, rows c-d).

**Ours vs. LDM.** In Section 2.1, we propose an alternative way to generate sketches by leveraging LDM (Rombach et al., 2022), as shown in Figure 3, row A. However, this approach deviates from the nature of sketching. Our proposed formulation, aligned with the topology of sketches (Figure 3, row C), resulted in significant improvements over LDM in both model complexity and performance, as depicted in Figure 13. Our formulation (Section 2.1) allows direct operation on the image space ($64 \times 64$) and compression of the Unet to a tiny variant without sacrificing performance. Despite the aggressive compression, our performance is significantly better than LDM, with respective Sketch-FID scores of 15.19 and 49, using a 41x smaller network.

**Sketch Intensity.** Controlling the intensity is crucial. Thus, we conducted further analysis, as shown in Table 3, that probes for best performance, the intensity should follow the data distribution. We refer to the intensity as the ratio between the white pixels to the black ones in the sketch. The reported FID

Table 3: Sketch Intensity Analysis.

| Dataset/Intensity | 19% | 24% | 31% | 37% | 43% |
|---|---|---|---|---|---|
| CelebHQ | **7.4** | 7.9 | 9.2 | 10.5 | 12.7 |
| Lsun-Churches | 7.8 | **6.9** | 8.3 | 8.9 | 10.1 |

Figure 14: Palette generation pipeline. First, we employ FastSAM (Zhao et al., 2023b) to segment the image. Then, we get the dominant color for each segment.

Figure 15: Ablation study for different input's types for the $2^{nd}$ stage on LSUN-Churches dataset (Yu et al., 2015).

Table 4: Comparison of the LDM concatenation mechanism against the schrödinger bridge (SB) using CelebHQ dataset.

Table 5: Training objective ablation study on CelebHQ dataset (Karras et al., 2017).

| | Input Type | FID $\downarrow$ |
|---|---|---|
| a) | Edges | 9.5 |
| b) | Edges + Palette-2 | 8.4 |
| c) | Sketch | 8.6 |
| d) | Sketch + Palette-1 | 8.3 |
| e) | Sketch + Palette-2 | **7.1** |
| f) | SAM (Colored) | 13.1 |
| g) | SAM (Edges) | 8.1 |

| $2^{nd}$ Stage | FID $\downarrow$ |
|---|---|
| Concat | 10.04 |
| SB | 9.45 |
| SB + Concat | **8.10** |

| Epochs | LDM | | Ours | |
|---|---|---|---|---|
| | $\epsilon$ | $x_0$ | $(x_t - x_0)$ | $x_0$ |
| 50 | 12.7 | 15.7 | 8.5 | 8.1 |
| 100 | 10.0 | 14.6 | 8.4 | 7.8 |
| 200 | 8.7 | 15.1 | 8.2 | **7.7** |

scores in Table 3 are the overall FID using all stages, and the intensity numbers are the initial intensity at $t = T$. The GT intensity for CelebHQ and Lsun-Churches are 19.2% and 23.5%, respectively.

## A.4 $2^{nd}$ STAGE: PALETTE

As depicted in Figure 14, we introduce a more realistic way to have high-quality palettes for free by first generating the segments from the RGB image using FastSAM (Zhao et al., 2023b). Then, a simple color detector module is utilized to get the dominant color, e.g., the median color per segment.

## A.5 TRAINING AND SAMPLING ALGORITHMS

---

**Algorithm 1** Training Pipeline

1: **for** $i = 1, \ldots, N$ **do**      ▷ Train each stage separately
2:     **for** $j = 1, \ldots, E$ **do**      ▷ Train for E epochs
3:        **for** $(x_0^i, y^i) \in \mathcal{D}$ **do**      ▷ Loop on the dataset $\mathcal{D}$
4:          $\epsilon \in \mathcal{N}(0, I)$      ▷ Sampling the noise
5:          $t \in U(1, \ldots, T)$      ▷ Sampling time-step
6:          Get $x_0^i$ by applying Eq. 3      ▷ $\tilde{x}_0$ based on the stage
7:          $x_t = \alpha_t x_0^i + (1 - \alpha_t) y^i + \sigma_t^2 \epsilon_t$      ▷ Forward process
8:          $\nabla_\theta \left\| \tilde{x}_0^i - p_\theta(x_0^i | x_t, y^i) \right\|^2$      ▷ Simplified objective

---

**Algorithm 2** Sampling Pipeline

1: $x_{inter} = []$      ▷ Initialize the list to save intermediate output for each stage
2: $y^1 = Zeros((H, W, 1))$      ▷ Initialize $1^{st}$ condition as a black image
3: $\epsilon \in \mathcal{N}(0, I)$      ▷ Sampling the noise once
4: **for** $i = 1, \ldots, N$ **do**      ▷ Sequential Sampling
5:     $x_T = y^i + \sigma_T^2 \epsilon_T$      ▷ Run forward process once
6:     **for** $t = T, \ldots, 1$ **do**      ▷ Progressive Sampling
7:        Get $x_{t-1}$ by applying Eq. 6 : $x_0^i = p_\theta(x_0 | x_t)$      ▷ Reverse process
8:     $y^{i+1} = x_0^i$      ▷ Update the condition for the next stage
9:     $x_{inter}.append(x_0^i)$      ▷ Store the intermediate outputs for each stage
10: **return** $x_{inter}$

---

## A.6 ABLATIONS

$1^{st}$ **Stage Representation.** In Section 2.2, we explore the versatility of the $3^{rd}$ stage (detailed image) by examining six input modalities, detailed in Figure 15. Comparing different contours representations, namely Edges (using Laplacian edge detector (Wang, 2007)), Sketch (utilizing PidiNet (Su et al., 2021)), and SAM-Edges (generated by SAM followed by Laplacian edge detector), we find that Sketch outperforms Edges, as edges tend to be noisier. However, SAM-Edges provides more detailed contours, yielding superior results. Notably, feeding SAM-Colored leads to significant performance degradation, likely due to color discrepancies, as observed in SAM in Figure 15. While SAM-Edges achieves optimal results, its computational intensity renders it impractical. In contrast, Sketch and Edges are computationally inexpensive. Furthermore, Sketch's sparse and user-friendly

nature makes it more suitable for editing, facilitating interpretation and modification than dense, noisy edges. Consequently, we adopt Sketch as the input modality for subsequent experiments.

**Palette Effect.** Adding more guidance will offer more editing abilities to the pipeline and enhance the performance. As shown in Figure 15, rows b and d, when we incorporate the palette into the contours, i.e., edges and sketch, the performance improved by almost 1 and 1.5, respectively. In addition, Palette-2 outperforms Palette-1 by 1 FID score. A more detailed analysis of fusing the stages together and how to mitigate the error accumulation can be found in Appendix A.7.

**Concatenation *v.s.* Schrödinger Bridge.** We formulate the generation problem as image-to-image translation. For instance, generating a palette given sketch or generating the final RGB image given overlaid palette and sketch. One possible approach is to utilize the conditioning mechanisms offered by LDM (Rombach et al., 2022), e.g., cross-attention and concatenation. However, these mechanisms are inefficient as their SNR tends to 0 as $t \to T$. In contrast, our formulation follows the Schrödinger bridge, which directly incorporates the condition $y$ into $x_t$ during the forward process. Accordingly, the SNR$\gg$0 at $t = T$, which allows reducing training steps from 1K to 50 (Figure 5) and achieving faster convergence (Figure. 4). Additionally, Table 4 demonstrates a quantitative comparison between Schrödinger Bridge and concatenation. As shown in Table, 4, the schrödinger bridge (SB) combined with the concatenation mechanism achieves the best results; however, SB alone is better than using the naive concatenation.

**Training Objectives.** Previous work (Rombach et al., 2022) showed that learning the noise $\epsilon$ directly enhances the performance compared to predicting the $x_0$. As, intuitively, predicting the difference $\epsilon_t \propto x_t - x_0$ is easier than predicting the original complicated $x_0$. However, this is not valid in our case due to the new term $y$, which makes predicting the difference much harder, as discussed in Section 2.1. Table 5 shows the same conclusion reported in (Rombach et al., 2022), that predicting the difference $\epsilon$ leads to better performance than $x_0$, where the FID score is 8.7 and 15.1, respectively. On the contrary, our method achieves almost the same performance regardless of the training objective, which is expected, as predicting the difference involves predicting $x_0$, Eq. 8.

## A.7   HOW TO FUSE DIFFERENT STAGES EFFICIENTLY?

Table 6: Systematic search for the best stopping step $s$ for the condition truncation.

| Metric/Steps ($s$) | 0 | 10 | 20 | 40 | 80 | 120 | 160 | 200 |
|---|---|---|---|---|---|---|---|---|
| $2^{nd}$ stage FID ↓ | **6.1** | 7.4 | 7.9 | 8.6 | 9.9 | 10.4 | 10.8 | 11.2 |
| Overall FID ↓ | 11.6 | 10.7 | **9.5** | 10.9 | 11.2 | 13.5 | 15.7 | 18.9 |

Table 7: Ablation study of the sketch augmentation effect on the overall performance after fusing the abstract and the detailed stages.

| | Cutout (DeVries & Taylor, 2017) Percentage | Dropout Percentage | $2^{nd}$ Stage FID ↓ | Overall FID ↓ |
|---|---|---|---|---|
| a) | 0 | 0 | 8.6 | 16.10 |
| b) | 5-10 | 5-20 | 9.89 | 13.94 |
| c) | 10-20 | 20-40 | 9.77 | 13.98 |
| d) | 20-30 | 50-70 | 9.80 | **13.76** |
| e) | 30-40 | 70-90 | 11.68 | 17.99 |

The reported FID scores for the $1^{st}$ and the $2^{nd}$ stages, in Figure 13 and Figure 15, respectively, are for each stage separately. In other words, in Figure 15, row c, the $2^{nd}$ stage achieves an 8.6 FID score when a GT sketch is fed, which is obtained from PidiNet (Su et al., 2021). However, when we fed the generated sketch from the $1^{st}$ stage, the performance drastically dropped from 8.6 to 16.1 (almost doubled), as shown in Table 7, row a, due to the domain gap between the generated and the GT sketches. To fill this gap, we explored two types of augmentations: 1) Cutout and Dropout augmentation. 2) Condition truncation augmentation.

**Cutout and Dropout Augmentation.** First, we explored the straightforward augmentation types, such as Cutout (DeVries & Taylor, 2017) and Dropout augmentations. For the Cutout, we apply a kernel to randomly blackout patches in the sketch. Additionally, regarding the dropout augmentation, we randomly convert white pixels to black pixels, interpreted as dropping some white points from the sketch. As shown in Table 7, generally, augmentation leads to a drop in the $2^{nd}$ stage performance while helping fill the gap in the overall performance, as the FID score decreased from 16 to almost 14. However, as shown in rows b-d, gradually increasing the amount of the applied augmentation does not help much, as the performance remains almost the same. However, the overall accuracy, i.e., FID, drops significantly from 14 to 18 when an aggressive augmentation is applied.

**Condition Truncation Augmentation.** As shown in Table 7, the conventional augmentation techniques do not help much. Accordingly, we explored another augmentation variant tailored for the

diffusion models, i.e., condition truncation (Ho et al., 2022). During training the $2^{nd}$ stage, we apply a random Gaussian noise to the fed condition, i.e., the sketch. So now the $2^{nd}$ stage is trained on noisy sketches instead of pure ones, which makes it more robust to the variations between the real sketches and the generated ones from the $1^{st}$ stage. Typically, we progressively generate the sketch in $T$ time steps; $T \to 0$. However, This added noise could be interpreted as we stop the sketch generation ($1^{st}$ stage) at a particular step $s$; $T \to s$. Consequently, we search for it during sampling to determine which step $s$ works best for the overall performance, as shown in Table 6. In other words, the $2^{nd}$ stage is trained on a wide range of perturbed sketches, e.g., pure sketch ($s = 0$), noisy one ($0 < s < T$), and even pure noise ($s = T$). The $1^{st}$ stage is trained for 200 steps, so $s = 200$ means we omit the sketch and feed pure noise. In contrast, $s = 0$ indicates that we feed the generated sketch as it is to the $2^{n}d$ stage. As shown in Table 6, following this truncation mechanism, the fusion of the two stages is performed more efficiently, where the overall performance drops from 16.1 to 10.6.

## A.8   CONDITIONING MECHANISM ERROR VULNERABILITY

In this section, we will compare the two conditioning mechanisms, Concatenation (Concat) and Schrödinger Bridge (SB), in terms of the error vulnerability.

While SB+Concat shows superior performance in Table 4 for conditional generation—when the input condition $y$ is the ground truth (GT)—its performance diminishes in the unconditional setup.

In the unconditional setup, $y$ is generated from the previous stage rather than being a perfect GT. This introduces errors into the generated $y$, leading to error accumulation, as discussed in Appendix A.7. Consequently, the model's performance suffers when relying on concatenation for conditioning. This is intuitive as $y$ is fed to the network in each denoising step $t$. Thus, the network emphasizes the $y$ a lot, making it more vulnerable to error accumulation.

To address this, we use only the Schrödinger Bridge (SB) as the conditioning mechanism for unconditional generation because of its demonstrated robustness against accumulated errors.

To this end, we tested a compromise approach by applying concatenation for 50% of the steps and then dropping the condition from concatenation by feeding zeros. The experiment is conducted on the LSUN-Churches dataset. Table 9 shows that this trick improves performance compared to concatenation in all steps. However, it still performs worse than using SB alone on generated sketches.

Accordingly, the choice of y-feeding strategy depends on the setup:

- Unconditional generation: Use only the Schrödinger Bridge, as it exhibits superior robustness against noise in the input condition.
- Conditional generation: Use a combination of Concatenation and Schrödinger Bridge, as shown in Table 3, for optimal results.

Table 8: Human evaluation for the real image editing capabilities between our method and SEED-X on 100 randomly sampled images from CelebHQ dataset.

| Method | FID |
| --- | --- |
| SEED-X | 2.6 |
| Ours | **4.3** |

Table 9: Comparison between different conditioning mechanisms regarding error vulnerability on the LSUN dataset.

| Method | FID |
| --- | --- |
| SB | **6.6** |
| SB+Concat (100% steps) | 10.1 |
| SB+Concat (50% steps) | 8.9 |

## A.9   FUTURE WORK

In this section, we will discuss some possible future work that shows the versatility of ToddlerDiffusion and its potential to drive innovation across image, video, and 3D tasks.

**Distill the knowledge from substage features for perception tasks.** The intermediate outputs from Toddler Diffusion (e.g., the sketch stage) can be reused as strong geometric priors for perception tasks by distilling the knowledge into perception networks to enforce geometric or perceptual constraints, making them structure-aware and geometry-guided aligned (Zhao et al., 2023a).

**Reusing Diffusion Features as High- and Low-Frequency Representations.** Toddler Diffusion's staged outputs inherently capture distinct frequency information, with the sketch stage emphasizing

low-frequency global structures and geometry and the palette and RGB stages capturing high-frequency details like textures. These features can be repurposed into a dual-branch network (Bi et al.; Wei et al., 2024): one branch focusing on low-frequency representations for tasks requiring global context (e.g., semantic segmentation, object localization) and the other specializing in high-frequency details for tasks like texture classification, instance segmentation, or boundary refinement. Integrating this knowledge into VLMs can improve performance and generalization for perception tasks.

**Addressing Limitations in T2I Models.** A significant challenge for text-to-image (T2I) models lies in handling complex prompts involving counting or spatial relations (Bakr et al., 2023; Ghosh et al., 2024). Fixing these issues in the sketch or abstract stage will be easier and simpler due to its lightweight. In addition, the subsequent stages (e.g., sketch-to-RGB) will remain unaffected and reusable. This approach allows targeted improvements without retraining/finetuning the entire pipeline, significantly reducing computational costs.

**Modular Prompt Handling for T2I Models.** Toddler Diffusion can simplify complex text prompts by decomposing them into modular components mapped to specific stages. For example, a detailed prompt can be split into geometry-focused instructions for the sketch stage and color-specific details for the sketch-to-palette stage. In addition, we can add more stages and specific prompts such as depth, e.g., $obj_1$ closer than $obj_2$. This enables a more focused and interpretable generation, particularly for long, detailed prompts with intricate requirements.

For instance: Given a complex prompt: "A bustling city street during sunset with a red double-decker bus on the right, a blue car parked on the left, people walking on the sidewalk, and sunlight casting long shadows on the ground." will be split into:

- Structure-Focused prompt: "A city street with a double-decker bus on the right, a car parked on the left, and people walking on the sidewalk."
- Palette-Focused prompt: "The double-decker bus is red, the car is blue, the sky shows a sunset with shadows on the ground."

**Video Generation in Abstract Domains.** Generating videos directly in the RGB domain is computationally expensive and prone to inconsistencies. Toddler Diffusion can split the problem into geometry consistency and stylistic consistency. Specifically, it first generates videos in an abstract domain (e.g., sketches), ensuring geometric consistency across frames. Then, these sketches can be refined into RGB frames using another stage, Sketch-to-RGB, conditioned on a reference frame for color and style consistency. This two-step process simplifies the complex task of maintaining temporal and stylistic coherence in video generation.

**Spatial Decompositionality.** An intriguing future direction is to adapt Toddler Diffusion to cascade generation across spatial dimensions instead of modalities:

- Object-Level Cascading: Generate foreground objects first, followed by backgrounds, enabling fine-grained control for game development or storyboarding applications.
- Part-Level Cascading: Decompose objects into parts (e.g., car wheels, body, interior) for industries like automotive design and industrial manufacturing, where precise control at the component level is critical.

**Enhancing 3D Generation.** Toddler Diffusion can streamline 3D generation by reusing the sketch stage for geometry, followed by a depth estimation stage, and finally, lifting 2D geometry into full 3D representations. This staged approach is particularly valuable for AR/VR scene creation, autonomous driving simulations, etc. Additionally, we can first generate the layout, followed by another stage that generates the meshes (Koo et al., 2023).

**Integration with NeRFs.** The sketch stage can provide strong structural priors for Neural Radiance Fields (NeRFs), improving 3D scene reconstruction from sparse views. This integration could accelerate NeRF optimization and enhance generalization in applications like scene editing and AR/VR reconstruction, where structural integrity is critical.

**Intrinsic Image Decomposition.** Toddler Diffusion's staged approach can be extended to tasks like intrinsic image decomposition, where images are disentangled into reflectance, illumination, and shading components. This decomposition enables applications like relighting, material editing, and realistic AR/VR rendering, where physically consistent outputs are crucial.

## B    MORE QUALITATIVE RESULTS

### B.1    DEMO

Please refer to our *demo*, which probes our controllability capabilities in performing consistent edits.

Our demo demonstrates a lot of use cases, including:

- (0:00 –> 0:20) [1st stage] Starting from noise generates a sketch by running only the 1st stage, which is unconditional generation.
- (0:20 –> 0:24) [2nd stage] Starting from a generated sketch generates a palette. This one is considered an unconditional generation as noise generates the sketch unconditionally.
- (0:24 –> 0:32) [3rd stage] Starting from a generated palette generates an RGB image. This one is also considered an unconditional generation.
- (0:32 –> 0:44) [Conditional Generation] Starting from the GT sketch, we generate an RGB image.
- (0:44 –> 1:15) [Editing] We show our sketch-based editing capabilities.
- (1:15 –> 1:46) [Editing] We show our palette-based editing capabilities.

### B.2    REAL IMAGE EDITING

As explained in Section 2.4, we ensure consistent edits for real images by storing the noise used during the generation process for later edits. The primary challenge with real image editing is that we don't have access to the noise values at each denoising step, as our model didn't generate the image. To address this, we employ DDPM-inversion (Huberman-Spiegelglas et al., 2024) to reconstruct the intermediate noise. As illustrated in Figure 16, we compare our model against SEED-X (Ge et al., 2024), and our model achieves consistent edits on real reference images better than SEED-X, despite being trained on much fewer data compared to SEED-X and using smaller architecture. Additionally, SEED-X fails to perform the edits correctly and keeps the rest of the image unedited, such as the rest of the face or the background. On the contrary, our method shows great faithfulness to the original image while performing the edits depicted in the sketch. We have created a small test-set for editing where we manually designed 100 edits on CelebHQ and generated the corresponding edits for both our method and SEED-X. Then, we asked annotators (4 annotators) to give a score based on the aestheticism of the generated image and the adherence to the edit command. As shown in Table 8, the annotators scores show great preferences to our method over SEED-X where our score is 4.3 while SEED-X is only 2.6

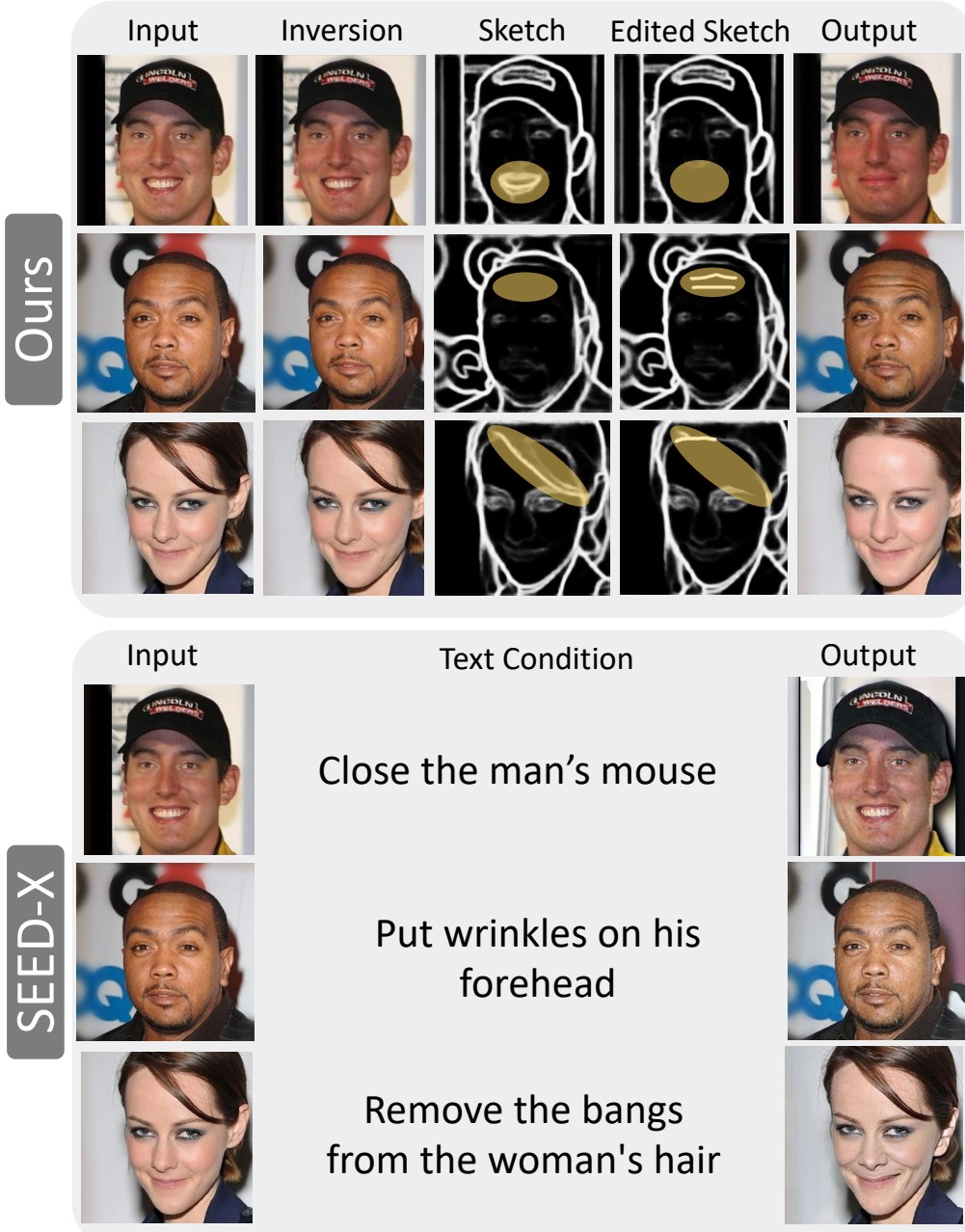

Figure 16: Comparison between our approach, ToddlerDiffusion and SEED-X (Ge et al., 2024) in terms of the real images capabilities.

## B.3 CLASS-LABEL ROBUSTNESS

Beyond these quality comparisons, we also conducted a robustness analysis to assess our model's behavior when faced with inconsistencies between conditions. Specifically, we deliberately provided incorrect class labels for the same sketch. This mismatched the sketch (geometry) and the desired class label (style). Despite the inconsistent conditions, as illustrated by the off-diagonal images in Figure 17, our model consistently adheres to the geometry defined by the sketch while attempting to match the style of the misassigned category, demonstrating the robustness of our approach. Successful

cases are highlighted in yellow, and even in failure cases (highlighted in red), our model avoids catastrophic errors, never hallucinating by completely ignoring either the geometry or the style.

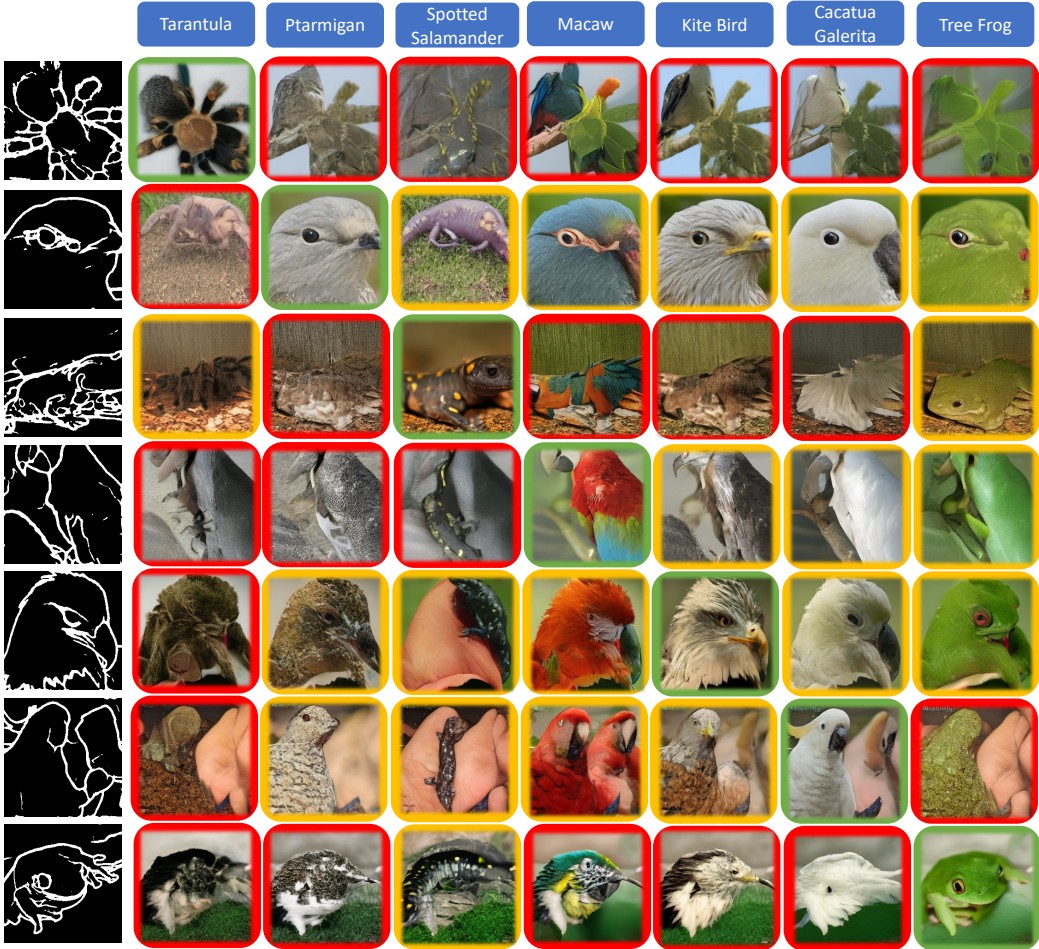

Figure 17: Our results on the ImageNet dataset. The first column depicts the input sketch. The diagonal (green) represents the generated output that follows the sketch and the GT label. The off-diagonal images show the output given the sketch and inconsistent label. These adversarial labels show our model's robustness, where the successful cases are in yellow, and the failure ones are in red.

## B.4 EDITING CAPABILITIES

Our approach introduces an interpretable generation pipeline by decomposing the complex RGB generation task into a series of interpretable stages, inspired by the human generation system (Stevens, 2012; Marr, 2010; 1974). Unlike traditional models that generate the complete image in one complex stage, we break it into $N$ simpler stages, starting with abstract contours, then an abstract palette, and concluding with the detailed RGB image. This decomposition not only enhances interpretability but also facilitates dynamic user interaction, offering unprecedented editing capabilities for unconditional generation, as shown in Figure 18, Figure 19, Figure 20, and Figure 21.

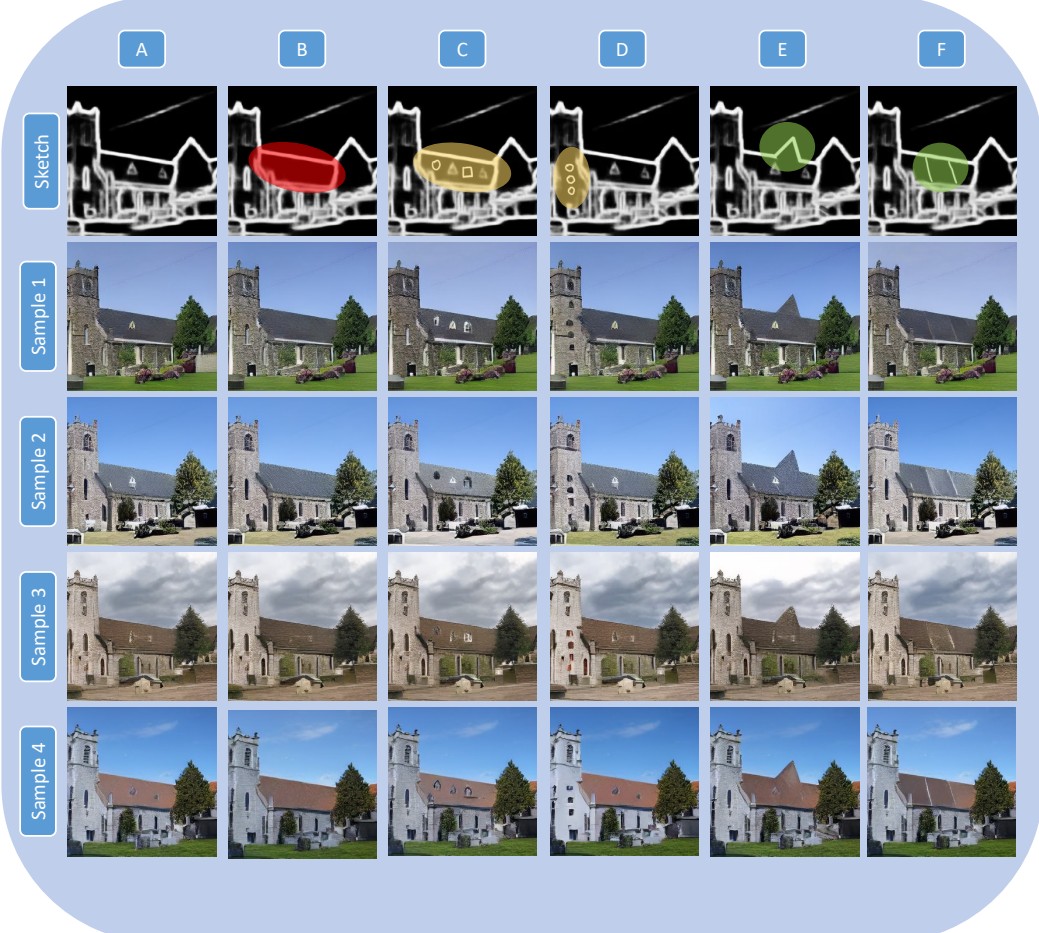

Figure 18: Controllability ability of our framework, ToddlerDiffusion. Starting from generated sketch and RGB image (A), we can remove artifacts or undesired parts, in red, (B), add a new content, in yellow, (C-D), and edit the existing content, in green, (E-F) by manipulating the sketch.

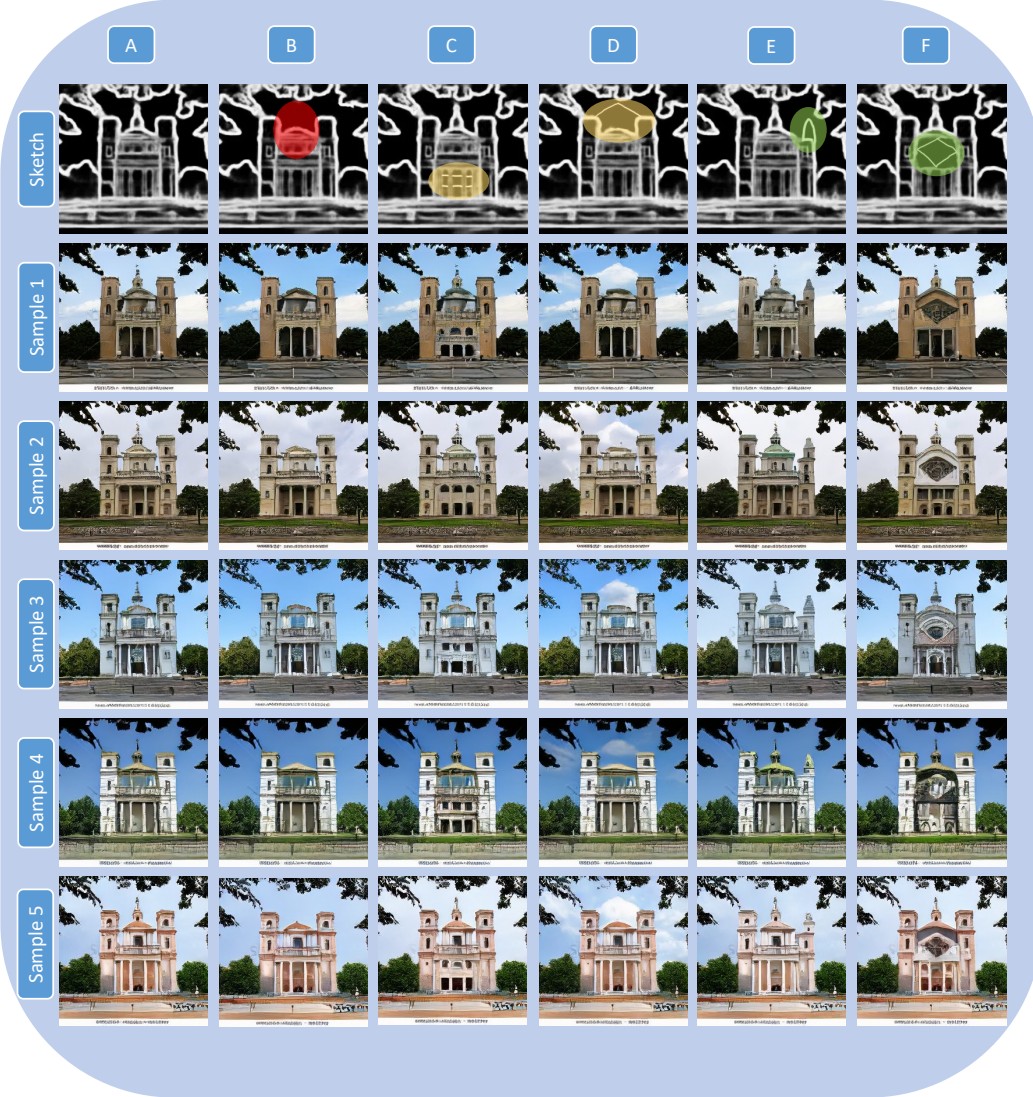

Figure 19: Controllability ability of our framework, ToddlerDiffusion. Starting from generated sketch and RGB image (A), we can remove artifacts or undesired parts, in red, (B), add a new content, in yellow, (C-D), and edit the existing content, in green, (E-F) by manipulating the sketch.

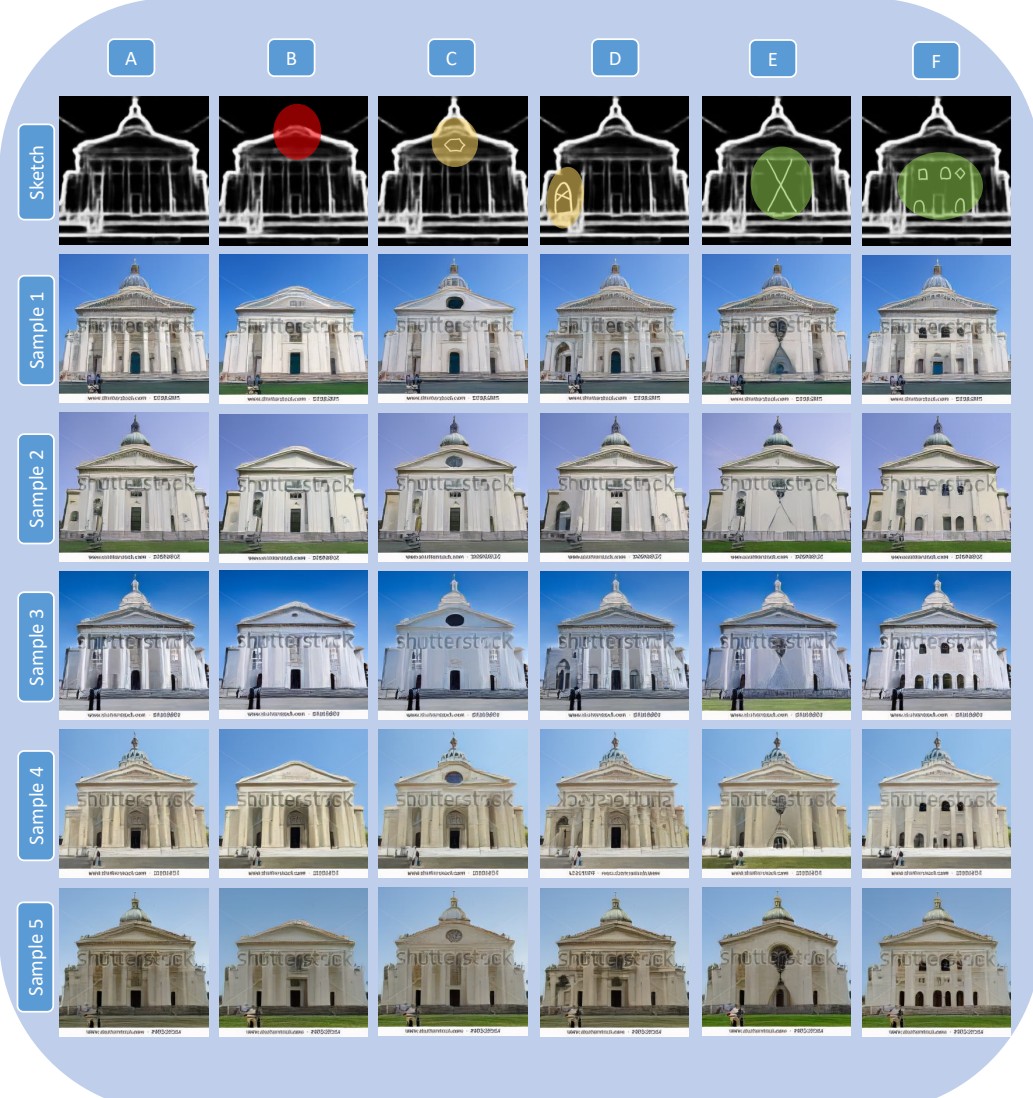

Figure 20: Controllability ability of our framework, ToddlerDiffusion. Starting from generated sketch and RGB image (A), we can remove artifacts or undesired parts, in red, (B), add a new content, in yellow, (C-D), and edit the existing content, in green, (E-F) by manipulating the sketch.

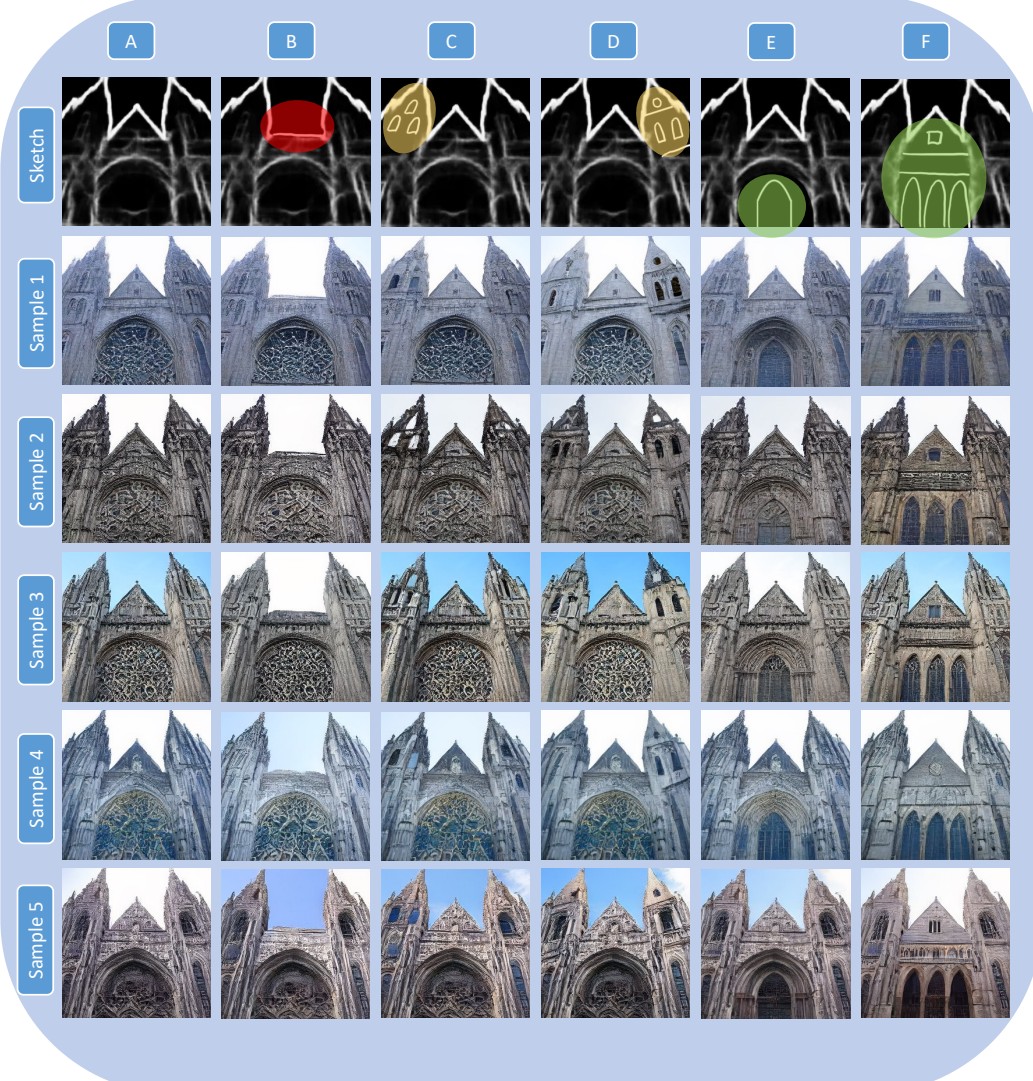

Figure 21: Controllability ability of our framework, ToddlerDiffusion. Starting from generated sketch and RGB image (A), we can remove artifacts or undesired parts, in red, (B), add a new content, in yellow, (C-D), and edit the existing content, in green, (E-F) by manipulating the sketch.

### B.5 SKETCH CONDITIONING (SD-v1.5)

We convert our method to a "sketch2RGB" by omitting the $1^{st}$ stage to show our controllability capabilities. By doing this, we can compare our model against the vast tailored sketch-based models built on top of SD-v1.5. However, these methods have been trained for a long time on large-scale datasets. Thus, for a fair comparison, we retrain them using CelebHQ datasets for 5 epochs starting from SD-v1.5 weights. As demonstrated in Figure 22, despite this is not our main focus, our method outperforms conditioning methods without adding additional modules (Zhang et al., 2023; Zhao et al., 2024) or adapters (Mou et al., 2024). We compare training-free (Tumanyan et al., 2023; Parmar et al., 2023) and training-based (Li et al., 2023b; Zhang et al., 2023; Zhao et al., 2024) methods. The performance is worse than expected for the trained-based methods due to two reasons: 1) We train all the models for only 5 epochs; thus, some of this method, in contrast to us, needs more epochs to converge. 2) We apply augmentations to the input sketch, which shows that some of these methods are vulnerable to sketch perturbations.

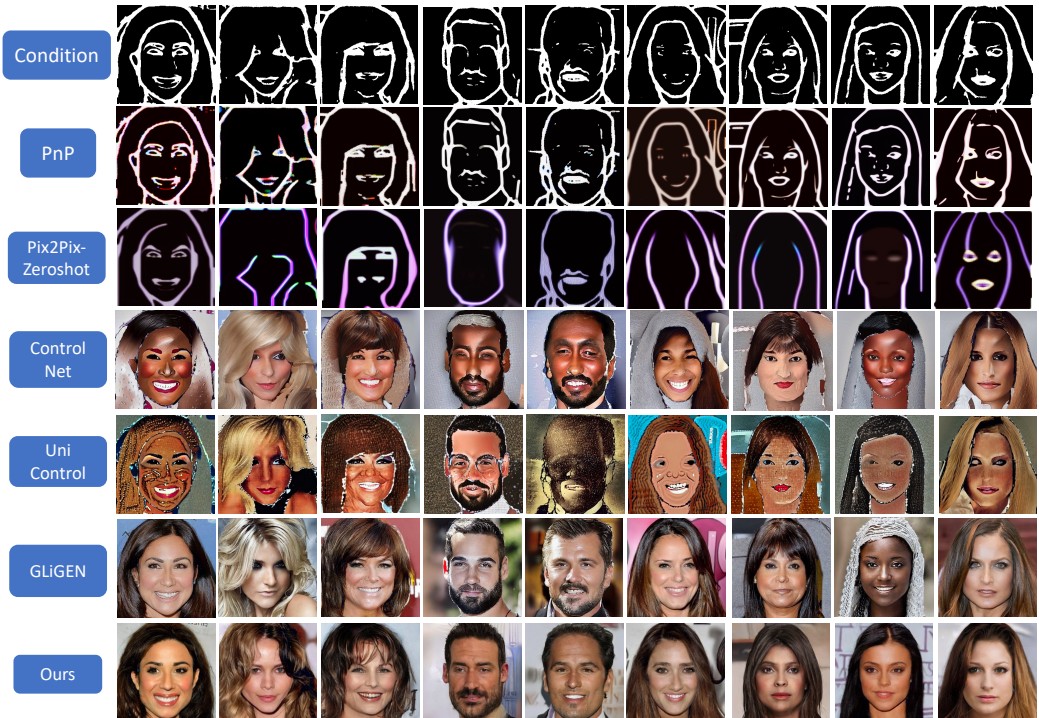

Figure 22: Comparison between our approach, ToddlerDiffusion and training-free (Tumanyan et al., 2023; Parmar et al., 2023) and training-based (Li et al., 2023b; Zhang et al., 2023; Zhao et al., 2024) sketch conditioning methods.

## B.6   TEXT + SKETCH CONDITIONING (SD-V1.5)

**CelebHQ:** We conducted another experiments to condition the model on both text and sketch. To this end, we have extended the CelebHQ dataset by adding a detailed text description for each image using Llava-Next v1.6. As Llava generates too long descriptions, we ask GPT-4o to summarize them. We train the model starting from the SDv1.5 weights for only five epochs. As shown in Figure 23, we feed the same sketch with different prompts, where the only difference between them is the gender, then the model generates the RGB images that follow both the sketch and the text conditioning.

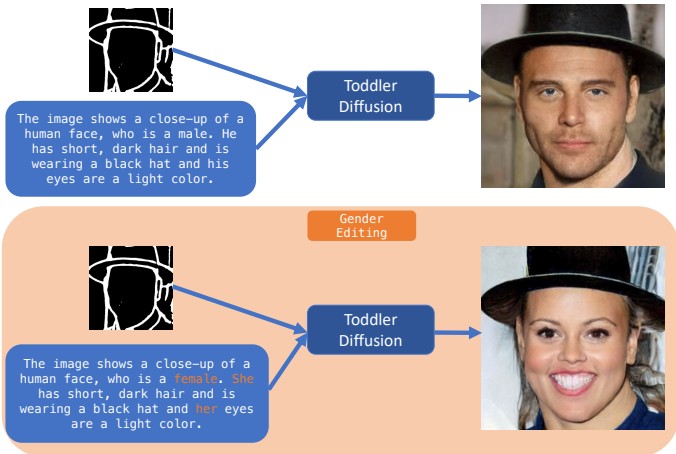

Figure 23: Text and Sketch conditioning capabilities of our proposed method.

**LionArt:** Furthermore, we conducted a more challenging experiment to condition the model on both text and sketch on a more challenging and diverse dataset, LionArt. To this end, we randomly sample 1 M images from LionArt and train our model for only 1 epoch starting from SDv1.5 weights. As shown in Figure 24, our model generates RGB images that adheres to both conditions; the text prompt and the input sketch.

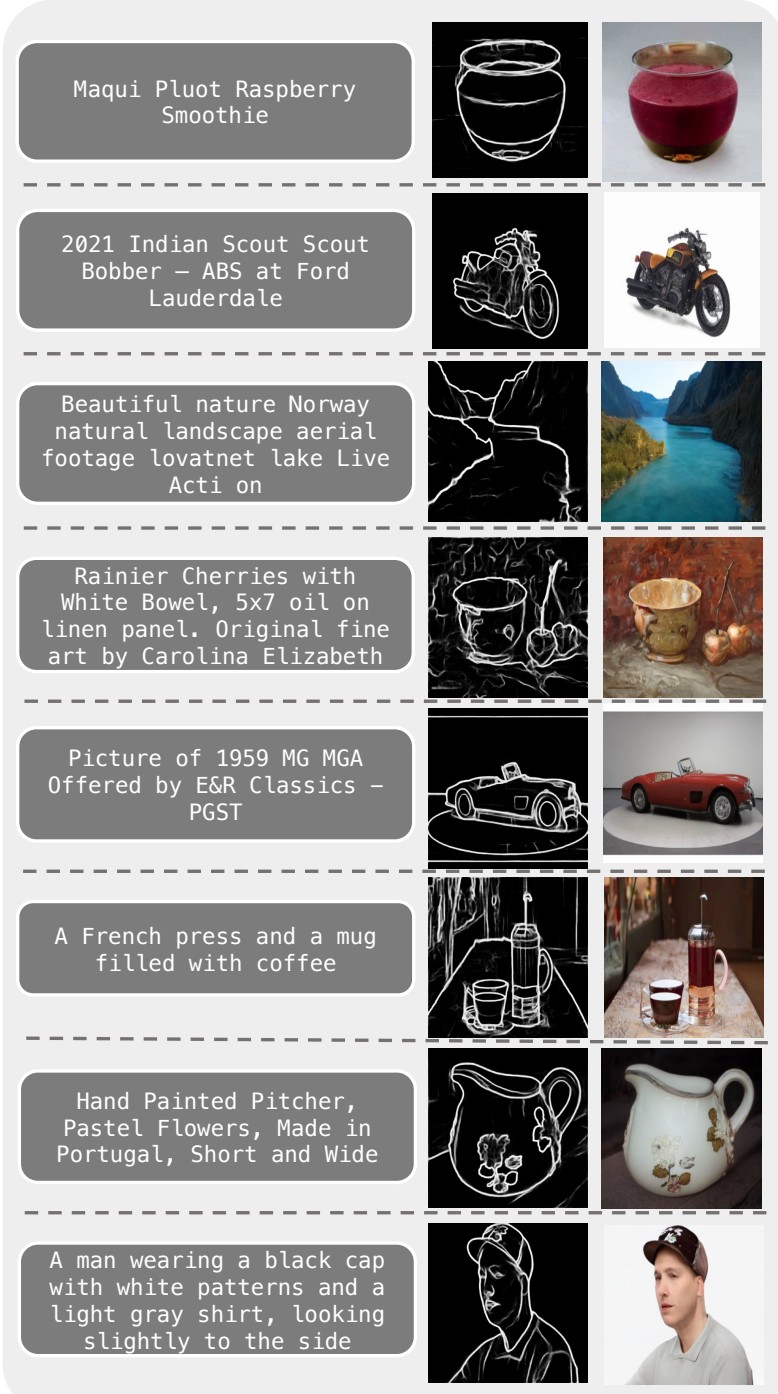

Figure 24: Qualitative results for our Text-conditional generation on LionArt dataset.

## B.7 TEXT CONDITIONING

To show the versatility of our approach, we conducted experiments to show our capabilities in generating sketches from text on two different datasets: CUB200 and Dress Code dataset. The Caltech-UCSD Birds-200-2011 (CUB-200-2011) dataset is the most widely-used dataset for fine-grained visual categorization task. It contains 11,788 images of 200 subcategories belonging to birds. Due to its importance, DM-GAN extends the dataset to include a text description for each image. There is no GT paired sketches for each image, thus, we run PidiNet to generate sketch for each RGB image. Additionally, we show our capabilities on the Dress Code dataset, which is more than 3x larger than publicly available datasets for image-based virtual try-on and features high-resolution paired images (1024x768) with front-view, full-body reference models. Like CUB200, the Dress Code data is an unimodal dataset that only contains RGB images. Fortunately, Ti-MGD extends the dataset to be a multi-modal dataset by having image-text pairs.

In this setup we have two conditioning mechanisms: 1) The sketch $y$, which is handled by the forward process (Eq.2). 2) Cross-attention layer to handle the text condition.

We have trained the $1^{st}$ stage for 1K epochs as it is very small, only 5 M parameters, and the dataset scale is very small, thus the 1k epochs takes less than 12 hours using a single A100 GPU. Then, we train the $2^{nd}$ stage, 141 Million parameters, for only 200 epochs. In contrast, LDM takes longer to converges as shown in Figure 4, thus we train it for 400 epochs.

As shown in Table 10 and Table 11, we have achieved better accuracies on both datasets compared to LDM despite being training for half the epochs. For qualitative results, please refer to Figures 25 and 26.

Table 10: Comparison between our approach and LDM using text conditioning multi-modal dataset, Dress Code (Baldrati et al., 2024).

| Method | FID |
|--------|-----|
| LDM    | 10.7 |
| Ours   | **9.1** |

Table 11: Comparison between our approach and LDM using text conditioning multi-modal dataset, CUB200 (Zhu et al., 2019).

| Method | FID |
|--------|-----|
| LDM    | 12.2 |
| Ours   | **10.8** |

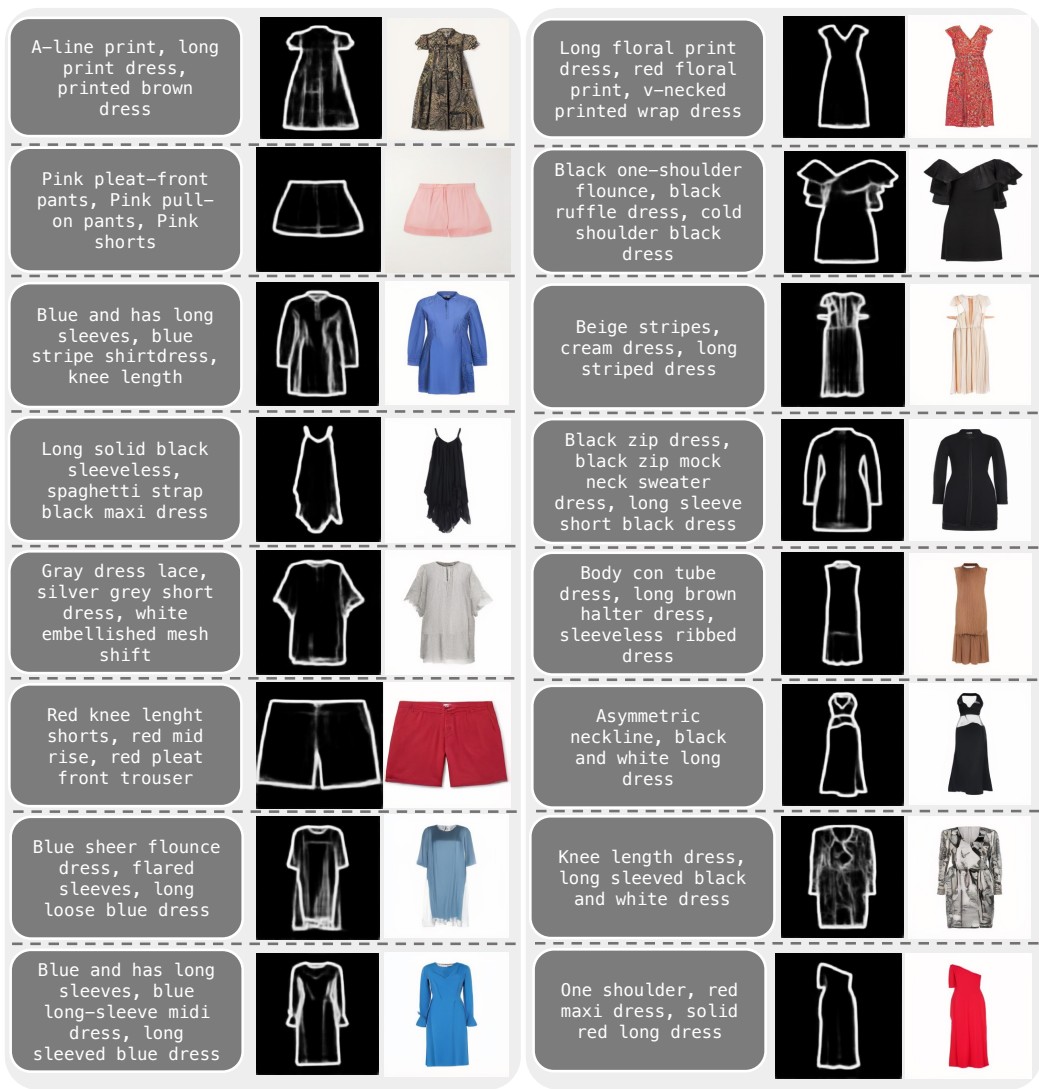

Figure 25: Qualitative results for our Text-conditional generation on Dress Code dataset (Baldrati et al., 2024). The first column is the input text prompt, second column is the generated sketch, last column is the generated RGB image.

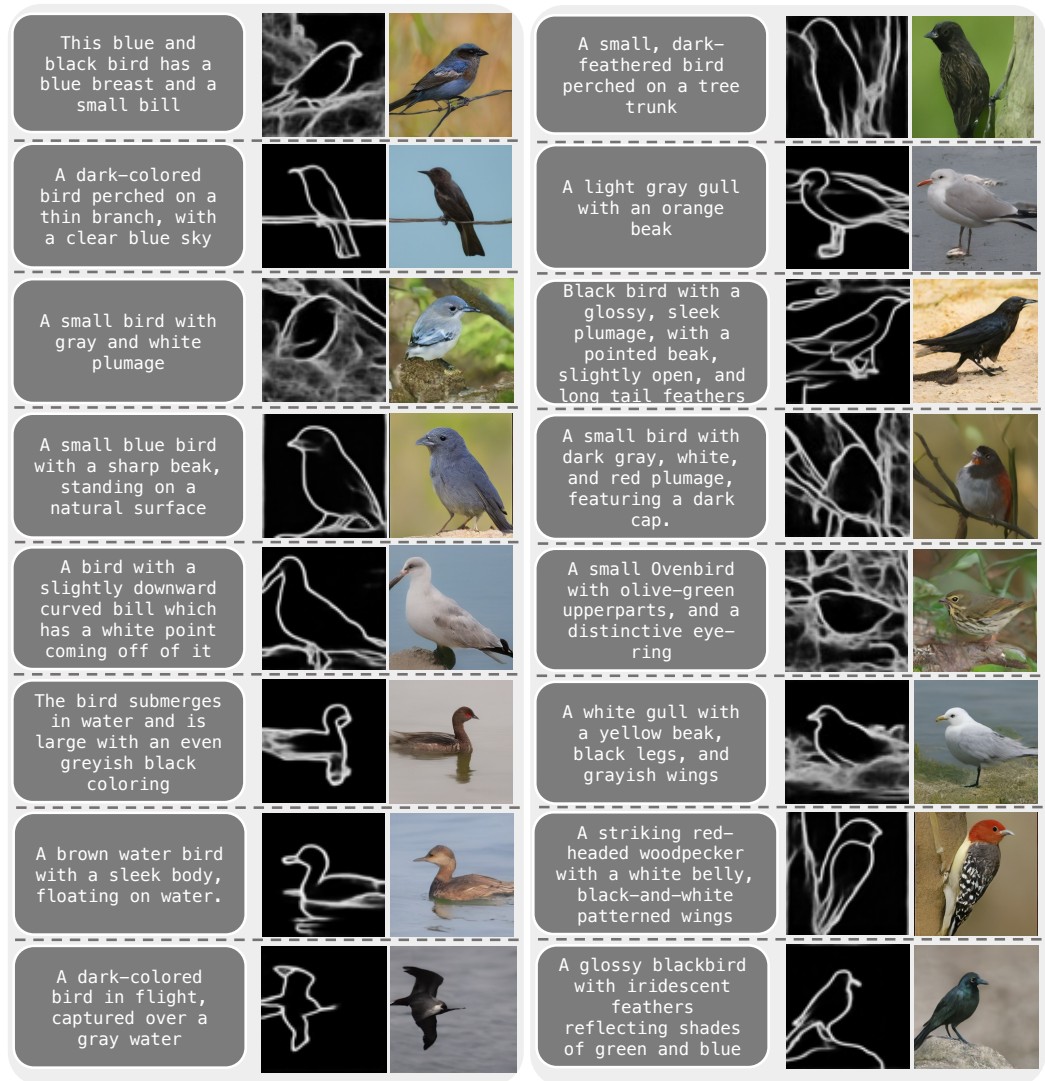

Figure 26: Qualitative results for our Text-conditional generation on CUB200 dataset (Zhu et al., 2019). The first column is the input text prompt, second column is the generated sketch, last column is the generated RGB image.

## B.8 FREE-HAND SKETCH CONDITIONAL GENERATION SAMPLES

Figure 27 depicts more qualitative results for our conditional generation pipeline, where we first generate the sketch using drawing tools and then use the generated sketch to generate the RGB image. Despite the sketch being too bad and sparse, our model still generates high-quality images.

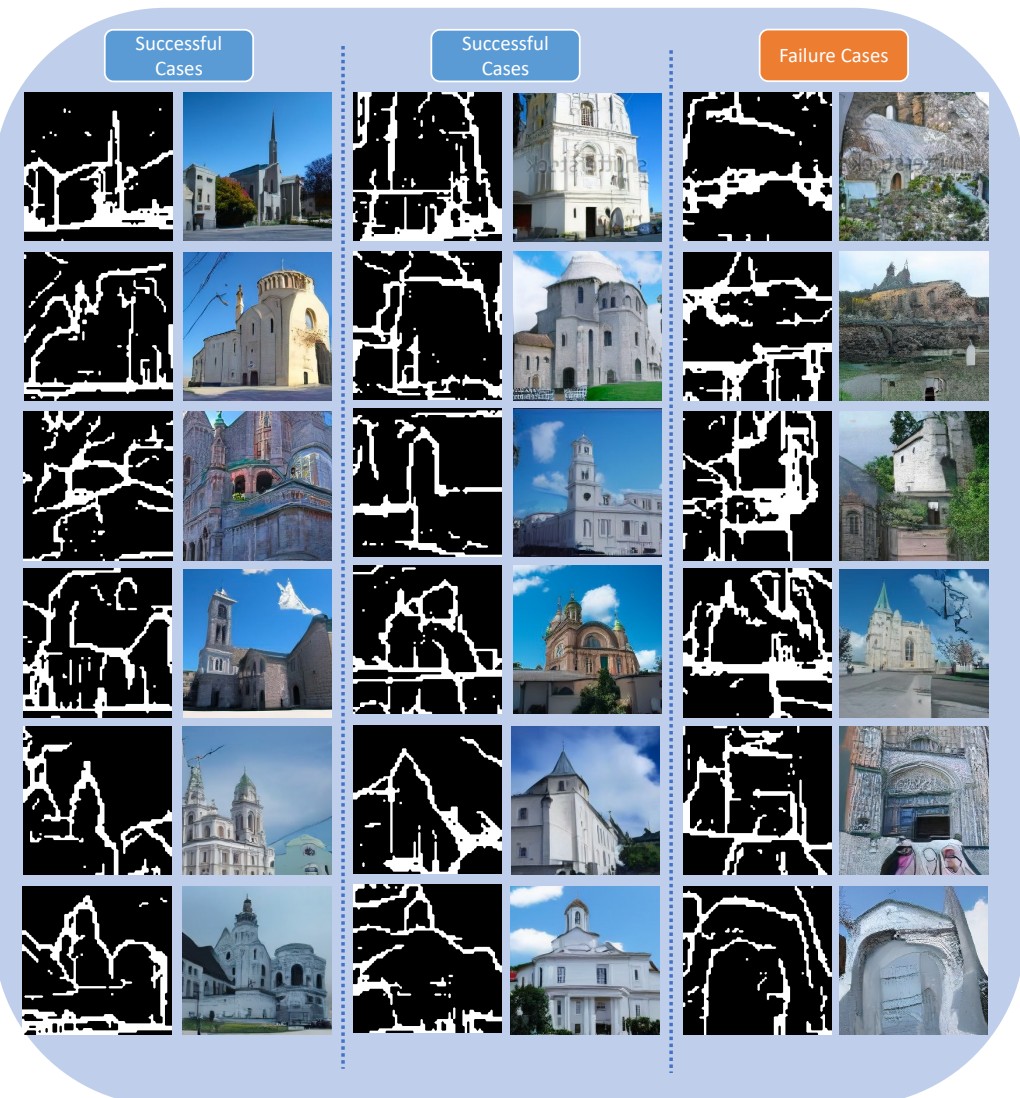

Figure 27: Qualitative results for our unconditional generation.

### B.9 QUALITATIVE RESULTS FOR OUR SKETCH-CONDITIONAL GENERATION

In addition to our unconditional generation, our framework can be used as a conditional generation pipeline, where the user can omit the first stage and directly fed a reference sketch. As shown in Figure 28, given the reference sketch, we run only the second stage, which is responsible for generating the RGB image given a sketch. Compared to the generated images in Figure 27, the generated images in Figure 28 are much better as the reference sketches are much better than the generated ones. This observation is aliened with our discussion regarding the gap between the two stages. Therefore, improving the quality of the generated sketches will directly reflect on the overall images quality.

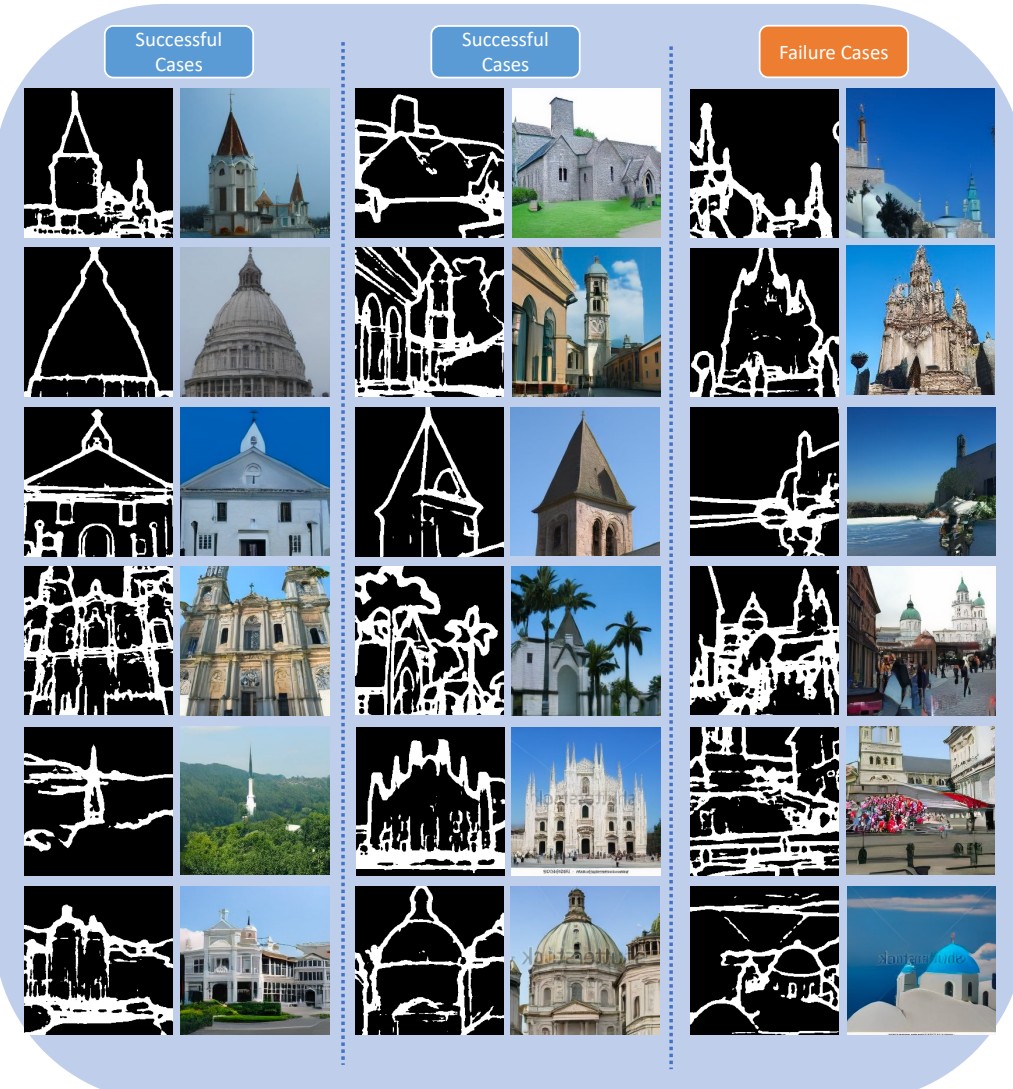

Figure 28: Qualitative results for our sketch-conditional generation.

## C  FAILURE CASES

The last column of Figure 27 and Figure 28 depicts failure cases, where the generated RGB is following the sketch but generates unrealistic image. Intuitively, the failure cases are more apparent in the unconditional generation due to the generated sketches quality.

