# OpenReview forum: "ToddlerDiffusion: Interactive Structured Image Generation with Cascaded Schrödinger Bridge"
_ICLR.cc/2025/Conference — ICLR 2025 Poster_

### Official Review · Reviewer_byq3 · 2024-11-03

**Soundness:** 3
**Presentation:** 3
**Contribution:** 3
**Rating:** 6
**Confidence:** 4

**Summary:**

This paper proposes a new approach to decompose the diffusion framework into simpler and interpretable stages. The new framework called ToddlerDiffusion is a cascaded model that introduces two additional intermediate representations, sketch and palette. Different modality-specific models are connected to obtain high-quality image outputs with detailed textures. Instead of naive LDM, the Schrödinger Bridge is leveraged to determine the optimal transformation between modalities. The new design achieves better overall performance with faster sampling and smaller architecture compared with LDM, and also enables editing and interaction at intermediate steps.

**Strengths:**

1.	The framework natively offers decent capability in editing and interpretability without additional adapters or fine-tuning.
2.	By decomposing the image generation into three simpler sub-tasks, the framework has fewer parameters, converges faster, and is more robust against a smaller number of sampling steps.
3.	The framework shows its capacity to achieve better performance on given datasets comparing with naive LDM.

**Weaknesses:**

1.	There is a large gap between the performance of the baseline model (LDM/SD1.5) and the current state-of-the-art models.
2.	The experiments showed the model’s capability in limited training data and resources, however, the value of proving a smaller model converges faster is limited. There’s lack of the evidence about its performance compared with baselines given sufficient data and training.
3.	The description of the experiments lacks many details that reduced the robustness of the conclusions. For example, the experiments comparing editing capabilities with controlnet did not mention which base model was used by controlnet or at which resolution was the experiment performed, makes the fairness of the experiment questionable.

**Questions:**

1.	Given ToddlerDiffusion is a smaller framework compared with LDM, showing it converges faster and has better performance with the same number of epochs is not sufficient to prove it’s a better model. Table 2 claims the training converges at epoch 400 (not mentioning on which dataset), but in Table 1 none of the datapoints was reported at epoch 400. Showing that ToddlerDiffusion outperforms when both training procedures converge would be a stronger evidence.
2.	Please provide more details about the experiments, such as but not limited to: at which resolutions were the three stages trained and evaluated? How was controlnet/SDEdit configured when the editing capacities was compared? Is there any quantitative results supporting it’s editing capacity?
3.	It’s understandable that The generation quality of ToddlerDiffusion is inferior to that of the SOTA model (SDXL, Flux.1, etc.) given limited data and training. However, The use cases for low-resolution unconditional or label-conditioned generation are very limited. Would the first two stages be able to handle more complicated conditions at higher resolutions as in latest models? If so, would the framework still be efficient?

---

> ### Author Response · Authors · 2024-11-22
> **Response 1, Part 1**
>
> ---
> > **Q1: What will be the performance compared with baselines given sufficient data and training?**
>
> > **Q1.1: Table 2 claims the training converges at epoch 400 (not mentioning on which dataset), but in Table 1 none of the data points was reported at epoch 400.**
>
> You are correct, and we appreciate your attention to detail. In Table 2, the training actually converges at epoch 600 (not 400). We have fixed this typo in the revised version.
>
> The experiments in Table 2 were conducted on the CelebHQ dataset.
>
> > **Q1.2: Showing that ToddlerDiffusion outperforms when both training procedures converge would be stronger evidence.**
>
> The number of epochs reported in Table 1 follows the training procedure followed by LDM, ensuring a fair comparison. Accordingly, the results reported in Table 1 on the three datasets represent the performance after convergence.
>
> Moreover, as illustrated in Figure 4, our approach converges significantly faster than LDM, further highlighting its efficiency and effectiveness.
>
> ---
> > **Q2: [Experiments Details] Please provide more details about the experiments**
> > **Q2.1: which resolutions were the three stages trained and evaluated?**
>
> * **First two stages:**
>
> The first two stages operate at a low resolution (64×64) in the pixel space rather than the latent space, ensuring efficiency during training and evaluation.
>
> * **Third Stage:**
>
> The third stage follows the LDM configuration. Specifically, we use VQ-GAN-f4 to encode and decode the RGB image:
>
> * The input RGB image (256×256×3) is first encoded into a latent representation (64×64×3).
>
> * The U-Net is trained in the latent space, and the resulting latent $z_0$ is then decoded back to the original resolution to obtain the final output $x_0$ (256×256×3).
>
> For a detailed flow, please refer to Appendix A2 and Figures 12 and 13.
>
> > **Q2.2: How was controlnet/SDEdit configured when the editing capacities was compared?**
>
> * **[ControlNet]**
>
> For fairness, we followed the training configuration provided in the ControlNet source code. Specifically:
>
> * We start from the converged model train on the CelebHQ dataset for 600 epochs.
>
> * We train our method and ControlNet for an additional 50 epochs with the sketch as a condition.
>
> * The input resolution for training was set to 256×256×3, consistent with the LDM configuration.
>
> * **[SDEdit]**
>
> SDEdit enables zero-shot editing capabilities and, therefore, struggles to condition sketches effectively, as shown in Figure 7. Its capabilities are thus limited primarily to palette conditioning.
>
> * We used the standard configuration detailed in the SDEdit source code.
>
> * The parameter $t_0$ was set to 0.5, where $t_0=0$ represents the guide itself, and $t_0=1$ indicates a random sample.
>
> ---
> > **Q3: Is there any quantitative results supporting the editing capacity?**
>
> To evaluate the editing capacity, we created a small test set for editing by manually designing 100 edits on the CelebHQ dataset. We generated corresponding edits using both our method and SEED-X.
>
> For evaluation, we asked 4 annotators to score the results based on two criteria:
>
> 1. Aesthetic Quality: The visual appeal of the generated images.
>
> 2. Adherence to the Edit Command: How well the generated image aligns with the specified edit.
>
> As shown in the table below, the annotators' scores indicate a strong preference for our method over SEED-X. Specifically, our method achieved an average score of 4.3, compared to SEED-X's score of 2.6.
>
> These results and further details have been updated in Appendix B.2.
>
> | **Method** | **Score** |
> |------------|---------|
> | SEED-X     | 2.6     |
> | Ours       | **4.3**     |

---

> ### Author Response · Authors · 2024-11-22
> **Response 1, Part 2**
>
> ---
> > **Q4: Would the first two stages be able to handle more complicated conditions at higher resolutions as in latest models? If so, would the framework still be efficient?**
>
> To demonstrate the versatility of our approach and its ability to handle more complex conditions (e.g., text conditioning), we conducted experiments on two datasets: CUB200 and Dress Code (see Appendix B.7).
>
> **[CUB200 and Dress Code]**
>
> **CUB200 Dataset:**
>
> The Caltech-UCSD Birds-200-2011 (CUB200) dataset is widely used for fine-grained visual categorization tasks. It contains 11,788 images across 200 bird subcategories. To enable text-to-sketch generation, we extended the dataset using the text descriptions provided by DM-GAN. Since the dataset lacks ground truth (GT) paired sketches, we employed PidiNet to generate sketches for each RGB image.
>
> **Dress Code Dataset:**
>
> The Dress Code dataset is a large-scale, high-resolution dataset (1024x768) with paired front-view, full-body reference models, primarily used for image-based virtual try-on tasks. While the dataset itself is unimodal (RGB images), Ti-MGD extended it to include image-text pairs, enabling multimodal experiments.
>
> **Conditioning Mechanisms:**
>
> In these experiments, we employed two conditioning mechanisms:
>
> 1. The sketch $y$ is handled by the forward process (Equation 2).
> 2. A cross-attention layer to manage the text conditioning.
>
> **Training Details:**
>
> * The first stage has only 5M parameters. Given the small model size and dataset scale, we trained it for 1K epochs, taking less than 12 hours on a single A100 GPU.
>
> * The second stage has 141M parameters. We trained it for 200 epochs.
>
> For comparison, LDM took longer to converge, as shown in Figure 4. Consequently, we trained LDM for 400 epochs for a fair comparison.
>
> **Results:**
>
> As shown in the tables below, our approach achieved superior performance on both datasets with fewer training epochs:
>
> | **Dress Code** | **FID** |
> |------------|---------|
> | LDM        | 10.7    |
> | Ours       | **9.1**   |
>
> | **CUB200** | **FID** |
> |------------|---------|
> | LDM        | 12.2    |
> | Ours       | **10.8**   |
>
> For qualitative results, please refer to Figures 25 and 26.
>
> **[LionArt]**
>
> Furthermore, we conducted a more challenging experiment to condition the model on both text and sketch on a more challenging and diverse dataset, LionArt.
> To this end, we randomly sample 1 M images from LionArt and train our model for only 1 epoch starting from SDv1.5 weights.
> As shown in Figure 24, our model generates RGB images that adhere to both conditions: the text prompt and the input sketch.

---

> > ### Comment · Reviewer_byq3 · 2024-11-29
> >
> > Thank for the detailed response from the authors, I am raising my rating by 1pt.

---

> > > ### Author Response · Authors · 2024-12-01
> > > **Thanks**
> > >
> > > * We sincerely thank you for your valuable feedback, the engaging discussion, and for raising your rating.
> > >
> > > * As we still have time until December 3rd, we would be happy to address any additional concerns or questions you may have to further clarify or enhance your assessment of our work.
> > >
> > > * Please feel free to share your thoughts, and we will do our best to address them promptly.

---

### Official Review · Reviewer_rf6t · 2024-11-03

**Soundness:** 3
**Presentation:** 4
**Contribution:** 2
**Rating:** 8
**Confidence:** 4

**Summary:**

The paper introduces Toddler diffusion, a method to generate images like the thought process of a child. The paper decomposes the image generation process into three stages first on a high level sketch generation, then a colour palette generation and finally generating a complete RGB image. To enable learning from one modality to another, the authors leverage Schrödinger Bridges. Such a form a training enables capabilities of seamless editing into the model. Moreover also enables faster sampling compared to conventional LDMs. The authors perform comparisons with conventional LDMs to illustrate the effectiveness of the method.

**Strengths:**

1. The proposed framework of a multistep generation process is novel to the best of my knowledge. The method is motivated by an inspiring thought process based on the workflow of how humans think to create a picture.
2. The disentangled training process also enables enhanced controllability, stability and faster training and inference process. Moreover such a generation process also gets benefits of added diversity because of the compositional effect in each stage
3. The performance boost over naive LDMs is substantial while also enabling zero shot capabilities to the the model.
4. The method achieves better performance than LDMs across diverse datasets while showing better training and inference efficiency.

**Weaknesses:**

1. The training process of toddler diffusion is not clear from the main manuscript. Are different U-Nets leveraged from different modality transformations. I would advise the authors to include a flowchart diagram of signal flow in the main manuscript or alternatively add a subsection detailing this aspect.
2. Toddler diffusion utilizes Schrödinger bridges and an SDE perspective for the training and sampling process. However Rectified Flows have shown to achieve much better performance utilizing ODEs, which also improves further distillation performances. Can the authors contrast on the benefits of training and sampling using diffusion  Schrödinger bridges and rectified flows.
3. Toddler diffusion requires annotated data in different modalities to enable conditional generation. Hence I believe the comparisons provided by the authors by comparing with Naive LDMs are not fair, for a fairer evaluation. I believe the authors need to train a diffusion model which can generate sketch from noise, then 2 conditional models for the translation tasks. Could the authors please provide a comparison with LDMs trained in the proposed manner or alternatively comment on how the current experiments are valid comparisons
4. Can the authors also include performance of Toddler diffusion compared to pixel space diffusion architectures and diffusion transformers for the same datasets. Some valid methods would be SiT[1], DiT[2], Pixel space models[3,4]

   [1] Ma, N., Goldstein, M., Albergo, M.S., Boffi, N.M., Vanden-Eijnden, E. and Xie, S., 2024. Sit: Exploring flow and diffusion-based generative models with scalable interpolant transformers. arXiv preprint arXiv:2401.08740.

   [2] Peebles, W. and Xie, S., 2023. Scalable diffusion models with transformers. In Proceedings of the IEEE/CVF International Conference on Computer Vision (pp. 4195-4205).

   [3] Kingma, D. and Gao, R., 2024. Understanding diffusion objectives as the elbo with simple data augmentation. Advances in Neural Information Processing Systems, 36.

   [4] Kynkäänniemi, T., Aittala, M., Karras, T., Laine, S., Aila, T. and Lehtinen, J., 2024. Applying guidance in a limited interval improves sample and distribution quality in diffusion models. arXiv preprint arXiv:2404.07724.

**Questions:**

1. Minor spelling mistake Ln 46  emerging proprieties

2. Regarding the problem prescribed in section 2.1 Ln 216,217 : [1] provides some suggestions on how to schedule the noise process to achieve zero terminal SNR and fix the inference schedule. I refer the paper here for the reference of the authors

   [1] Lin, S., Liu, B., Li, J. and Yang, X., 2024. Common diffusion noise schedules and sample steps are flawed. In Proceedings of the IEEE/CVF winter conference on applications of computer vision (pp. 5404-5411).

---

> ### Author Response · Authors · 2024-11-22
> **Response 1, Part 1**
>
> ---
> > Q1: [Training Process]
>
> > Q1.1: Are different U-Nets leveraged from different modality transformations?
>
> Yes, we use three different U-Nets, and the complexity of each is detailed in Table 2.
>
> Our main hypothesis, inspired by toddler growth, is that decomposing a complex problem into simpler sub-problems allows us to address each smaller problem with a more efficient architecture and achieve faster convergence.
> This approach is exemplified in Figure 14 (right table), where we use a U-Net architecture that is 98x smaller than LDM while still generating better sketches.
>
> Moreover, as shown in Table 2, our overall architecture remains faster and more efficient than LDM despite utilizing three U-Nets across the stages.
> This is because we use smaller, task-specific models for each stage, which aligns directly with our core hypothesis.
>
> > Q1.2: Include a flowchart diagram of the signal flow.
>
> Please refer to Appendix A.2, specifically Figure 12 and Figure 13, for the detailed flow.
>
> ---
> > Q2: Can the authors contrast the benefits of training and sampling using diffusion Schrödinger bridges and rectified flows?
>
> We appreciate the reviewer’s question regarding comparing diffusion Schrödinger bridges (SB) and rectified flows. Both methods address similar goals in generative modeling but have distinct advantages depending on the use case.
>
> Rectified Flows (RFs) define the forward process as straight paths between the
> data distribution and a standard normal distribution:
>
> $x_t = (1-t) x_0 + t \epsilon$
>
> Accordingly, RF uses a deterministic sampling procedure, avoiding the need for stochastic diffusion paths.
>
> In contrast, the SB formulation directly aligns the source and target distributions by solving the stochastic control problem, making it highly suitable for conditional generation tasks.
>
> $x_t = (1-\alpha_t) x_0 + \alpha_t y + \sigma^2 \epsilon$
>
> SB simplifies the learning process by starting from a meaningful conditional domain (e.g., sketch or palette), reducing the reliance on large model capacities and extensive training data, as shown in our experiments (e.g., Figures 4 and 24).
>
> ---
> > Q3: [Fair Comparison]
> > I believe the comparisons with Naive LDMs are unfair. For a fair evaluation, the authors should train a diffusion model for sketch generation and two conditional models for translation tasks. Could the authors provide a comparison with LDMs trained this way?
>
> We acknowledge that there are multiple ways to implement the ToddlerDiffusion concept. One straightforward approach is to directly use the concatenation mechanism proposed in LDM. However, this approach fails to validate our hypothesis: by decomposing the complex generation task into simpler steps, inspired by a child’s growth, we can tackle these simpler tasks with a more efficient architecture.
>
> **[$1^{st}$ stage]**
>
> As shown in Figure 14 (right table), LDM fails to generate high-quality sketches when using a small architecture. In contrast, our approach, leveraging a tiny U-Net (1.9M parameters), achieves significantly better performance while being 98x smaller.
>
> The novel design of the $1^{st}$ stage is a key contribution to our paper. Specifically, starting from Gaussian noise (as in LDM) prevents the use of a tiny network. Instead, we start from a black canvas and progressively add noise at $t=T$, which serves as control points during the denoising steps.
>
> **[$3^{rd}$ stage]**
>
> For the $3^{rd}$ stage, which generates the final RGB images, we demonstrate in Table 3 that the Schrödinger Bridge (SB) achieves better performance compared to the concatenation mechanism.
>
> Additionally, Table 3 shows that leveraging both conditioning mechanisms (SB + concatenation) further improves performance, achieving an FID of 8.1 compared to 10 when using concatenation alone.
>
> **[Overall Performance]**
>
> The table below summarizes the performance of each stage and the overall framework.
>
> Despite experimenting with LDM variants, including increasing the network capacity for each stage, our method consistently outperforms LDM in all metrics while maintaining architectural efficiency.
>
> | **Method**   | **Noise-->Sketch** | **Noise-->Sketch** | **Sketch-->Palette** | **Sketch-->Palette** | **Palette-->RGB** | **Palette-->RGB** |
> |--------------|--------------------|--------------------|----------------------|----------------------|-------------------|-------------------|
> |              | # Param            | FID                | # Param              | FID                  | # Param           | FID               |
> | LDM (Small)  | 1.9 M              | 33.2               | 1.9 M                | 25.6                 | 101 M             | 10.5              |
> | LDM (Large)  | 187 M              | 23.5               | 187 M                | 11.6                 | 263 M             | 8.1               |
> | Ours (Small) | 1.9 M              | 15.1               | 1.9 M                | 9.6                  | 101 M             | 7.1               |

---

> ### Author Response · Authors · 2024-11-22
> **Response 1, Part 2**
>
> ---
> > Q4: [More Comparisons]
> > Can the authors also include the performance of Toddler diffusion compared to pixel space diffusion architectures and diffusion transformers for the same datasets?
>
> We appreciate the reviewer's suggestion.
> Before the rebuttal, we had already explored CDM [1] as another baseline besides LDM to reinforce our method.
> Fortunately, it was a pixel space method. Thus, we hope it will answer your concern, as the concept is exactly the same.
> However, if you prefer to try the mentioned references, please let us know, and we will do our best to incorporate our method into them.
>
> **[Pixel Space Diffusion]**
>
> The first two stages of ToddlerDiffusion—Noise to Sketch and Sketch to Palette—already operate in pixel space at a resolution of 64×64, making them efficient for lower-resolution tasks.
>
> For the $3^{rd}$ stage, we replaced LDM with the Cascaded Diffusion Model (CDM) [1], which operates in pixel space, first generating at smaller resolutions and then progressively increasing resolution to improve training efficiency.
>
> Experiments on CelebHQ Dataset:
>
> 1. Training the baseline (CDM) on the CelebHQ dataset.
>
> 2. Training only the $3^{rd}$ stage in pixel space following the CDM approach, as the first two stages were already trained in pixel space.
>
> The results below demonstrate the superiority of our method, where we boost CDM’s performance by 1.4 FID points:
>
> | **Method** | **FID** |
> |------------|---------|
> | CDM        | 9.6     |
> | CDM+Ours   | **8.2**     |
>
> **[Transformer-based Diffusion]**
>
> We integrated the U-ViT architecture [2] into our approach and trained both the original U-ViT and our framework for 200 epochs on the CelebHQ dataset.
>
> As with other results in the paper (e.g., Figure 4 and Table 1), our method significantly boosts the performance of U-ViT while converging faster, as shown in the table below:
>
> | **Method** | **FID** |
> |------------|---------|
> | U-ViT        | 14.1     |
> | U-ViT+Ours   | **11.4**     |
>
> [1] Ho, Jonathan, et al. "Cascaded diffusion models for high fidelity image generation." Journal of Machine Learning Research 23.47 (2022): 1-33.
>
> [2] Bao, Fan, et al. "All are worth words: A vit backbone for diffusion models." Proceedings of the IEEE/CVF conference on computer vision and pattern recognition. 2023.
>
> ---
> > Q5: Regarding the problem prescribed in section 2.1 Ln 216,217 : [1] provides some suggestions on how to schedule the noise process to achieve zero terminal SNR.
>
> We appreciate the reviewer’s suggestion and the reference to [1].
>
> To clarify, we mentioned the SNR analogy in a slightly different context. Specifically, our hypothesis—validated empirically in Figure 4 and Table 1—is that leveraging the Schrödinger Bridge to transport the conditional domain $y$ to the target domain $x$ facilitates the learning process, resulting in better performance and faster convergence.
>
> We relate this to the SNR analogy as follows:
>
> * In the standard unconditional generation process, the model starts from pure Gaussian noise (SNR \approx 0) and must learn a complex trajectory to reach the RGB domain.
>
> * In contrast, as shown in Figure 2, our approach simplifies the learning process by starting from a relevant conditional domain $y$, which acts as a shortcut. This results in a steady generation trajectory and an easier learning process.
>
> By starting from a meaningful conditional domain, we effectively reduce the difficulty of the generation task, improving both performance and convergence.

---

> > ### Comment · Reviewer_rf6t · 2024-11-25
> >
> > I thank the authors for the detailed response.
> >
> > [W1]. Thank you for including the query regarding the number of U-Nets involved and the signal flow diagram in the appendix. This clarifies to the readers the exact underlying process.
> > [W2]. The optimal transport perspective of rectified flows enable translation from Domain 1 to Domain 2 rather than noise to a particular domain. Hence the formulation will be like  Schrödinger bridges without the noise components. Rectified flows having portrayed lower FID scores and lower sampling times maybe suited for some parts of toddler diffusion. I think the authors should include this aspect in the future works section. Can the authors comment on whether there is any particular advantage using a stochastic process like  Schrödinger bridges over rectified flows.
> > [W3] I think the analysis provided by the authors on the FID scores of different LDMs confirm that  Schrödinger bridges works better than latent diffusion for translation tasks. It does not address my query of
> >  training conditional diffusion models.
> > [W4] I thank the authors for including the comparisons with pixel space and diffusion transformers. Please include these results in the appendix.

---

> > > ### Author Response · Authors · 2024-11-27
> > > **Response 2, Part 1**
> > >
> > > We thank the reviewer for their prompt response and valuable comments and feedback.
> > >
> > > ---
> > > > **W2: Comparison between Schrödinger Bridge and the Rectified flows.**
> > >
> > > We thank the reviewer for raising this insightful point.
> > >
> > > To address this, we are conducting a detailed analysis comparing Schrödinger Bridges and Rectified Flows in the context of conditional generation.
> > >
> > > Fortunately, the extended discussion period allows us to perform additional experiments and provide a comprehensive comparison.
> > >
> > > Once these experiments are complete, we will share our findings with you here.
> > >
> > > ----
> > > > **W3: I think the analysis provided by the authors on the FID scores of different LDMs confirms that Schrödinger bridges work better than latent diffusion for translation tasks. It does not address my query of training conditional diffusion models.**
> > >
> > > We are glad that our experiments and analysis confirm the effectiveness of Schrödinger Bridges over Latent Diffusion for translation tasks.
> > >
> > > Regarding your query about training conditional diffusion models, we may not have fully understood this aspect of your question. Can you please clarify it more?
> > >
> > > To clarify, we have implemented a more comprehensive baseline by comparing it against Conditional-LDM models.
> > >
> > > Specifically, we trained three types of LDM-based models:
> > >
> > > 1. Unconditional LDM: Trained to generate sketches starting from noise.
> > > 2. Conditional-LDM: Trained to take sketches as input and generate palettes.
> > > 3. Conditional-LDM: Trained to take sketches and palettes as input and generate the final RGB image.
> > >
> > > If there is a specific aspect of training conditional diffusion models that remains unclear, we would be happy to address it further.

---

> ### Comment · Reviewer_rf6t · 2024-11-27
>
> I thank the authors for addressing the queries.
>
> [W3] Earlier the training two conditional models part was unclear to me from the tables. It is much clearer now. I understand that the authors have performed the exact experiments I requested.
> [W2] I'm curious to see the results of W2.
>
> Regardless, I'm convinced of the workings of toddler diffusion. I believe there may be tasks involving image intrinsics other than Image generation where toddler diffusion can play a vital role.
>
> If the authors can add a section regarding this to the paper (in the future works section, portraying other possible tasks where toddler diffusion may be applied), I will change my rating to accept. One possibility I see is that toddler diffusion may disentangle a problem to multiple stages and the substages may be reused to solve another problem. But I'm interested to learn more about the authors thoughts

---

> ### Author Response · Authors · 2024-11-27
> **Response 3, Part 1 (Future Directions)**
>
> ---
> > **Q6: Future directions:**
>
> We sincerely thank the reviewer for the thoughtful suggestion regarding future applications of Toddler Diffusion. We have reflected on this and propose several exciting directions where Toddler Diffusion could play a pivotal role beyond its current scope.
>
> 1. **[Distill the knowledge from substage features for perception tasks]**
> The intermediate outputs from Toddler Diffusion (e.g., the sketch stage) can be reused as strong geometric priors for perception tasks by distilling the knowledge into perception networks to enforce geometric or perceptual constraints, making them structure-aware and geometry-guided aligned.
>
> 2. **[Reusing Diffusion Features as High- and Low-Frequency Representations]**
> Toddler Diffusion's staged outputs inherently capture distinct frequency information, with the sketch stage emphasizing low-frequency global structures and geometry and the palette and RGB stages capturing high-frequency details like textures. These features can be repurposed into a dual-branch network [1][2]: one branch focusing on low-frequency representations for tasks requiring global context (e.g., semantic segmentation, object localization) and the other specializing in high-frequency details for tasks like texture classification, instance segmentation, or boundary refinement. Integrating this knowledge into VLMs can improve performance and generalization for perception tasks.
>
> 3. **[Addressing Limitations in T2I Models]**
> A significant challenge for text-to-image (T2I) models lies in handling complex prompts involving counting or spatial relations [3][4].
> Fixing these issues in the sketch or abstract stage will be easier and simpler due to its lightweight.
> In addition, the subsequent stages (e.g., sketch-to-RGB) will remain unaffected and reusable.
> This approach allows targeted improvements without retraining/finetuning the entire pipeline, significantly reducing computational costs.
>
> 4. **[Modular Prompt Handling for T2I Models]**
> Toddler Diffusion can simplify complex text prompts by decomposing them into modular components mapped to specific stages. For example, a detailed prompt can be split into geometry-focused instructions for the sketch stage and color-specific details for the sketch-to-palette stage.
> In addition, we can add more stages and specific prompts such as depth, e.g., $obj_1$ closer than $obj_2$.
> This enables a more focused and interpretable generation, particularly for long, detailed prompts with intricate requirements.
>
> ```
> For instance:
> * Complex Prompt:
> "A bustling city street during sunset with a red double-decker bus on the right, a blue car parked on the left, people walking on the sidewalk, and sunlight casting long shadows on the ground."
> will be split into:
>
> * Structure-Focused Prompt:
> "A city street with a double-decker bus on the right, a car parked on the left, and people walking on the sidewalk."
>
> * Palette-Focused Prompt:
> "The double-decker bus is red, the car is blue, the sky shows a sunset with shadows on the ground."
> ```
>
> 5. **[Video Generation in Abstract Domains]**
> Generating videos directly in the RGB domain is computationally expensive and prone to inconsistencies. Toddler Diffusion can split the problem into geometry consistency and stylistic consistency.
> Specifically, it first generates videos in an abstract domain (e.g., sketches), ensuring geometric consistency across frames. Then, these sketches can be refined into RGB frames using another stage, Sketch-to-RGB, conditioned on a reference frame for color and style consistency. This two-step process simplifies the complex task of maintaining temporal and stylistic coherence in video generation.
>
> 6. **[Spatial Decompositionality]**
> An intriguing future direction is to adapt Toddler Diffusion to cascade generation across spatial dimensions instead of modalities.
>     * Object-Level Cascading: Generate foreground objects first, followed by backgrounds, enabling fine-grained control for game development or storyboarding applications.
>     * Part-Level Cascading: Decompose objects into parts (e.g., car wheels, body, interior) for industries like automotive design and industrial manufacturing, where precise control at the component level is critical.

---

> ### Author Response · Authors · 2024-11-27
> **Response 3, Part 2 (Future Directions)**
>
> 7. **[Enhancing 3D Generation]**
> Toddler Diffusion can streamline 3D generation by reusing the sketch stage for geometry, followed by a depth estimation stage, and finally, lifting 2D geometry into full 3D representations. This staged approach is particularly valuable for AR/VR scene creation, autonomous driving simulations, etc.
> Additionally, we can first generate the layout, followed by another stage that generates the meshes [5].
>
> 8. **[Integration with NeRFs]**
> The sketch stage can provide strong structural priors for Neural Radiance Fields (NeRFs), improving 3D scene reconstruction from sparse views. This integration could accelerate NeRF optimization and enhance generalization in applications like scene editing and AR/VR reconstruction, where structural integrity is critical.
>
> 9. **[Intrinsic Image Decomposition]**
> Toddler Diffusion’s staged approach can be extended to tasks like intrinsic image decomposition, where images are disentangled into reflectance, illumination, and shading components. This decomposition enables applications like relighting, material editing, and realistic AR/VR rendering, where physically consistent outputs are crucial.
>
>
> We believe these directions showcase the versatility of Toddler Diffusion and its potential to drive innovation across image, video, and 3D tasks. Thank you again for your thoughtful suggestion, which has inspired us to explore these exciting possibilities.
>
> [1] Bi, Qi, et al. "Learning Frequency-Adapted Vision Foundation Model for Domain Generalized Semantic Segmentation." The Thirty-eighth Annual Conference on Neural Information Processing Systems.
>
> [2] Wei, Zhixiang, et al. "Stronger Fewer & Superior: Harnessing Vision Foundation Models for Domain Generalized Semantic Segmentation." Proceedings of the IEEE/CVF Conference on Computer Vision and Pattern Recognition. 2024.
>
> [3] Bakr, Eslam Mohamed, et al. "Hrs-bench: Holistic, reliable and scalable benchmark for text-to-image models." Proceedings of the IEEE/CVF International Conference on Computer Vision. 2023.
>
> [4] Ghosh, Dhruba, Hannaneh Hajishirzi, and Ludwig Schmidt. "Geneval: An object-focused framework for evaluating text-to-image alignment." Advances in Neural Information Processing Systems 36 (2024).
>
> [5] Koo, Juil, et al. "Salad: Part-level latent diffusion for 3d shape generation and manipulation." Proceedings of the IEEE/CVF International Conference on Computer Vision. 2023.
>
> [6] Zhao, Wenliang, et al. "Unleashing text-to-image diffusion models for visual perception." Proceedings of the IEEE/CVF International Conference on Computer Vision. 2023.

---

> ### Author Response · Authors · 2024-11-30
> **Response 3, Part 3 (Schrödinger Bridge vs. Rectified flows)**
>
> > **W2: Comparison between Schrödinger Bridge and the Rectified flows.**
>
> We sincerely thank the reviewer for their insightful points, which have enriched our paper and motivated further comparisons.
>
> > **W2.1: Can the authors comment on whether there is any particular advantage to using a stochastic process like Schrödinger bridges over rectified flows?**
>
> The primary disadvantage of Rectified Flows (RF) lies in their limited editing and inversion capabilities, which are crucial for applications requiring consistency and control. Recent efforts, such as RF-prior [1] and others [2], attempt to improve these capabilities but still fall short compared to diffusion-based methods like DDIM inversion, as these methods rely mainly on information leakage from the source image.
> In contrast, as shown in Figures 10 and 16 of our paper, ToddlerDiffusion excels in consistent editing for both generated and real images, highlighting the utility of Schrödinger Bridges in such tasks.
>
> > **W2.2: Rectified flows may be suited for some parts of toddler diffusion.**
>
> We conducted extensive experiments on CelebHQ to evaluate RF and its adaptations within our pipeline:
> 1. Baseline Comparisons:
>     * Original RF [3]: Achieved an FID of 11.5 when trained unconditionally using the default setup [default setup](https://github.com/gnobitab/RectifiedFlow/blob/main/ImageGeneration/configs/rectified_flow/celeba_hq_pytorch_rf_gaussian.py). Testing the shared checkpoint yielded a similar FID of 11.
>     * Latent RF (InstaFlow) [4]: We adapted InstaFlow [4] into the same code base as the training code [is not publically available](https://github.com/gnobitab/InstaFlow/issues/28), and we got almost the same performance, with an 11.8 FID score.
>
> 2. Integration into Toddler Diffusion:
>     * RF conditioned on sketches showed faster convergence than starting from pure noise but underperformed Latent Diffusion Models (LDM) with an FID of 10.8 (vs. 7.1 for our method).
>
> 3. Recent Enhancements in RF:
>     * Emerging works, like those improving RF formulations and SNR schedulers [5], may address some of these limitations. However, incorporating enhanced RFs into our pipeline is left as promising future work.
>
> **[Observations on Noise Scheduling in Schrödinger Bridges]**
>
> Additionally, we want to link the comparison with RF with an interesting observation we have while using the Schrödinger Bridges.
> We want to answer from this ablation to what extent we can straighten the bridge without hurting the performance.
> Our experiments revealed the impact of the maximum uncertainty value in the noise schedule, Eq.2 (Figure 11).
>
> We found that both increasing and overly decreasing maximum noise harm performance. As shown in the table below, a balanced noise peak (e.g., 0.5) optimally preserves information from the initial domain while enabling sufficient stochastic exploration:
>
> | Max Noise | 16 | 8    | 2    | 0.5 | 0.25 | 0.128 | 0.05 |
> |-----------|----|------|------|-----|------|-------|------|
> | FID Score | 13 | 12.6 | 10.2 | 7.1 | 8.2  | 9.5   | 15   |
>
> We hypothesize that techniques such as fine-tuning or distillation could enable further straightening of the noise scheduler while preserving performance, an exciting area for future exploration.
>
> **[Conclusion]**
>
> While RF may offer certain advantages, particularly in convergence speed, Schrödinger Bridges provide superior performance in editing, inversion, and handling intermediate representations, making them well-suited for ToddlerDiffusion’s multi-stage design. We will continue exploring RF’s integration as part of future work, leveraging recent advancements to refine and expand our pipeline.
>
> Thank you again for this valuable discussion and for inspiring new avenues of research.
>
> ---
> **References:**
>
> [1] Rout, Litu, et al. "Semantic image inversion and editing using rectified stochastic differential equations." arXiv preprint arXiv:2410.10792 (2024).
>
> [2] Yang, Xiaofeng, et al. "Text-to-Image Rectified Flow as Plug-and-Play Priors." arXiv preprint arXiv:2406.03293 (2024).
>
> [3] Liu, Xingchao, Chengyue Gong, and Qiang Liu. "Flow straight and fast: Learning to generate and transfer data with rectified flow." arXiv preprint arXiv:2209.03003 (2022).
>
> [4] Liu, Xingchao, et al. "Instaflow: One step is enough for high-quality diffusion-based text-to-image generation." The Twelfth International Conference on Learning Representations. 2023.
>
> [5] Esser, Patrick, et al. "Scaling rectified flow transformers for high-resolution image synthesis." Forty-first International Conference on Machine Learning. 2024.

---

> > ### Author Response · Authors · 2024-12-01
> > **Kind reminder: We are looking forward to your reply**
> >
> > We appreciate your engagement and insightful point, which significantly contributes to the paper.
> >
> > We want to follow up on whether the future directions we shared and updated in Section 4 of the paper are satisfactory for you.
> >
> > In addition, we still have two days (till December 3rd), so if you have any further comments that will enhance your assessment of our work, please let us know.

---

> > > ### Comment · Reviewer_rf6t · 2024-12-02
> > >
> > > Dear Authors.
> > >
> > > Thanks for including the results with different noise levels.
> > >
> > > I also appreciate the authors efforts on including potential future works and how toddler diffusion can be utilized to simplify complex problems.
> > >
> > > Since the authors have addressed all my concerns and provided the requested additional experiments. I'm increasing my rating to accept.

---

### Official Review · Reviewer_dsQk · 2024-11-04

**Soundness:** 3
**Presentation:** 3
**Contribution:** 2
**Rating:** 6
**Confidence:** 4

**Summary:**

This paper introduces a generative framework comprising three stages: noise to sketch, sketch to palette, and palette to RGB images. The paper uses the Schrodinger bridge I2SB [1] to train these stages. This framework takes less training time and samples efficiently. Moreover, the architecture is smaller and has better training convergence. This framework could be used for editing and control generation.

[1]: I2SB: Image-to-Image Schrödinger Bridge

**Strengths:**

1. The paper is well-written and easy to follow. I particularly like the idea of the "toddler," which is motivated by how the child grows up and learns everything.
2. This training framework is more training/sampling efficient than unconditional diffusion generation. It could be used for editing and control generation since it comprises many conditional generative problems trained by Schrodinger Bridge.
3. The experiment is well-organized.

**Weaknesses:**

1. The methodology seems simple since it decomposes the generative problem into subsequent conditional generative problems. Each problem can be solved directly using LDM or Schrodinger Bridge, and the paper uses Schrodinger Bridge to solve them like I2SB. Therefore, this paper's novelty is limited since it is just like you apply I2SB directly to solve conditional generative problems.
2. The paper mentions extending the method to Stable Diffusion but omits stage 1. This is a meaningful editing and controllable generation problem. However, for the text-to-image generation, I doubt the ability to train stage 1 mapping from (noise and text) to sketch using the Schrodinger Bridge. The reason behind this is that text is a completely different modality with RGB, sketch, and palette. Therefore, mapping from (noise and text) to sketch is not easily solved by unconditional Schrodinger Bridge. Further, the architecture should be larger and specifically designed for that task. This is an interesting question that should be discussed in the paper to answer the framework's scalability. The author could try to train such (noise and text) to sketch models on small text-to-image datasets such as CUB-200 or COCO dataset. [2] (see this work for small text-to-image problem)
3. The paper should include citations for [1] since their methodology is similar to the paper method. I2SB uses the Schrodinger bridge for image-image problems, which are the same problem as your decomposed conditional generative problems. The only difference is they solve image superresolution/inpainting/deblurring instead of sketch-to-palette and palette-to-image.
4. Minor: Several typos at Eq. 3 (the right-hand equation label should be sketch and palette)

[1]: I2SB: Image-to-Image Schrödinger Bridge
[2]: Vector Quantized Diffusion Model for Text-to-Image Synthesis

**Questions:**

My biggest concern is about the novelty of this work since the framework could be considered as many conditional generative models. Secondly, I am not sure about the ability to extend this framework to text-to-image generation; this could limit the applicability of the proposed method.

---

> ### Author Response · Authors · 2024-11-22
> **Response 1, Part 1**
>
> ---
> > Q1: Novelty
>
> > Q1.1: My biggest concern is about the novelty of this work since the framework could be considered as many conditional generative models:
>
> We sincerely appreciate your interest in the concept of the "toddler," which is inspired by the way a child learns and grows.
>
> While it might seem that our framework could be viewed as simply stacking conditional generative models, achieving our goals and demonstrating the superior abilities of ToddlerDiffusion requires a fundamentally novel approach.
>
> The central research question of our paper is: Can we implement a smaller and more efficient diffusion architecture by decomposing the complex generation task into simpler steps, inspired by a child’s developmental process?
>
> To address this, we explored various implementation strategies, including directly adopting the concatenation mechanism proposed in Latent Diffusion Models (LDM). However, as demonstrated in the below table, this approach failed to validate our hypothesis and did not yield the desired gains.
>
> | **Method**   | **Noise-->Sketch** | **Noise-->Sketch** | **Sketch-->Palette** | **Sketch-->Palette** | **Palette-->RGB** | **Palette-->RGB** |
> |--------------|--------------------|--------------------|----------------------|----------------------|-------------------|-------------------|
> |              | # Param            | FID                | # Param              | FID                  | # Param           | FID               |
> | LDM (Small)  | 1.9 M              | 33.2               | 1.9 M                | 25.6                 | 101 M             | 10.5              |
> | LDM (Large)  | 187 M              | 23.5               | 187 M                | 11.6                 | 263 M             | 8.1               |
> | Ours (Small) | 1.9 M              | 15.1               | 1.9 M                | 9.6                  | 101 M             | 7.1               |
>
> Key Novel Contributions:
>
> 1. The Novel Design of the First Stage:
>
> * In Figure 14 (right table), we show that LDM fails to generate good sketches with a small architecture. In contrast, our approach, which utilizes a tiny U-Net (1.9M parameters), achieves significantly better performance while being 98x smaller.
>
> * The key insight is that starting from Gaussian noise, as in the standard LDM approach, prevents the use of such a small network. Instead, our novel design begins with a black canvas and progressively introduces noise at $t=T$, interpreted as control points during the denoising steps. This design is crucial for enabling the use of a tiny network in the first stage.
>
> 2. The Use of Schrödinger Bridge in the Third Stage:
>
> * For the third stage, which generates the final RGB images, we demonstrate in Table 3 that the Schrödinger Bridge (SB) performs better than the concatenation mechanism alone.
>
> * Additionally, Table 3 illustrates that leveraging both conditioning mechanisms (SB + concatenation) further improves performance, achieving an FID of 8.1 compared to 10 when using concatenation alone.
>
> In summary, the novelty of our work lies in the careful decomposition of the generation task and the design of efficient mechanisms tailored to each stage, resulting in a smaller, more efficient architecture that surpasses existing approaches.
>
> > Q1.2: It is just like you apply I2SB directly to solve conditional generative problems.
>
> **[Motivation]**
>
> While our work might seem similar to I2SB, we are motivated by a fundamentally different hypothesis and inspiration. Drawing from the concept of toddler growth, our approach decomposes a complex generation task into simpler sub-problems. This decomposition allows us to address each sub-problem using a more efficient architecture and achieve faster convergence.
> Additionally, our framework is the first to provide high-level interpretability during the generation process, enabling consistent editing capabilities even in unconditional generation. This is particularly important for editing artifacts effectively, which is not achievable with I2SB.
>
> **[Fewer Steps]**
>
> We are the first to analyze and demonstrate the advantages of using the Schrödinger Bridge in both training and sampling processes. Notably, as shown in Figures 5 and 6, our framework achieves high performance with only 50 training steps, without any significant drop in quality. This efficiency is a unique contribution of our method compared to existing approaches.
>
> **[Error Propagation]**
>
> Connecting the cascaded stages is not straightforward and requires careful design to ensure the framework works as intended. We have thoroughly detailed these challenges and discussed our solutions in Appendix A.6.

---

> ### Author Response · Authors · 2024-11-22
> **Response 1, Part 2**
>
> ---
> > Q2: I am not sure about the ability to extend this framework to text-to-image generation; this could limit the applicability of the proposed method.
>
> We agree with the reviewer that directly applying the Schrödinger Bridge to handle text conditioning presents significant challenges due to the large gap between the two domains (the bridge edges), making convergence difficult.
>
> However, to demonstrate the versatility of our approach, we conducted experiments showcasing our ability to generate sketches from text on two distinct datasets: CUB200 and Dress Code (see Appendix B.7).
>
> **[CUB200 and Dress Code]**
>
> **CUB200 Dataset:**
>
> The Caltech-UCSD Birds-200-2011 (CUB200) dataset is widely used for fine-grained visual categorization tasks. It contains 11,788 images across 200 bird subcategories. To enable text-to-sketch generation, we extended the dataset using the text descriptions provided by DM-GAN. Since the dataset lacks ground truth (GT) paired sketches, we employed PidiNet to generate sketches for each RGB image.
>
> **Dress Code Dataset:**
>
> The Dress Code dataset is a large-scale, high-resolution dataset (1024x768) with paired front-view, full-body reference models, primarily used for image-based virtual try-on tasks. While the dataset itself is unimodal (RGB images), Ti-MGD extended it to include image-text pairs, enabling multimodal experiments.
>
> **Conditioning Mechanisms:**
>
> In these experiments, we employed two conditioning mechanisms:
>
> 1. The sketch $y$ is handled by the forward process (Equation 2).
> 2. A cross-attention layer to manage the text conditioning.
>
> **Training Details:**
>
> * The first stage has only 5M parameters. Given the small model size and dataset scale, we trained it for 1K epochs, taking less than 12 hours on a single A100 GPU.
>
> * The second stage has 141M parameters. We trained it for 200 epochs.
>
> For comparison, LDM took longer to converge, as shown in Figure 4. Consequently, we trained LDM for 400 epochs for a fair comparison.
>
> **Results:**
>
> As shown in the tables below, our approach achieved superior performance on both datasets with fewer training epochs:
>
> | **Dress Code** | **FID** |
> |------------|---------|
> | LDM        | 10.7    |
> | Ours       | **9.1**   |
>
> | **CUB200** | **FID** |
> |------------|---------|
> | LDM        | 12.2    |
> | Ours       | **10.8**   |
>
> For qualitative results, please refer to Figures 25 and 26.
>
> **[CelebHQ Dataset]**
>
> To further validate our approach, we extended the CelebHQ dataset by adding detailed text descriptions for each image using Llava-Next v1.6 (Appendix B.6). Since Llava produces overly long descriptions, we used GPT-4o to summarize them into concise text prompts.
>
> We fine-tuned the model starting from SDv1.5 weights for only five epochs. As shown in Figure 23 (Appendix B.6), the model successfully generated RGB images that adhered to both the sketch and the text conditions. For example, feeding the same sketch with different prompts varying only by gender resulted in RGB images that accurately reflected both conditions.
>
> ---
> > Q3: Minor: Several typos at Eq. 3 (the right-hand equation label should be sketch and palette)
>
> We carefully reviewed Equation 3 and believe it is correct as written.
>
> In this equation, we demonstrate the input condition $y^i$ and the target $x^i_0$ for the second and third stages.
>
> As shown in the right-hand side of the equation, for the $2^{nd}$ and $3^{rd}$ stages, the original RGB image $x^{rgb}_0$ is fed to the corresponding functions, $F_s$ and $F_p$, to generate the condition $y^i$ for the "Palette" and "Detailed Image" stages, respectively.
>
> ---

---

> > ### Comment · Reviewer_dsQk · 2024-11-26
> >
> > Thank you for addressing my concern. I still think proposed method is somehow straightforward but the author show the improvement on text-to-image, indicating this method is scalable  and is worth to share with the community. Therefore, I will raise my score to 6. Good luck with your ICLR and please include several discussion about schrodinger bridge works like I2SB in related work.

---

> ### Author Response · Authors · 2024-11-28
> **Thanks and Final Comments on the Novelty**
>
> We sincerely thank you for your thoughtful feedback and kind words and for raising the score.
> Your acknowledgment of the scalability and utility of ToddlerDiffusion to the community is greatly appreciated.
>
> While we understand that novelty can be subjective, we appreciate your recognition of the value of our method to the field.
> As a final effort to convince you about our novelty, we would like to share two key points:
>
> 1. **[Straightforward Does Not Diminish Originality]:**
> While ToddlerDiffusion may seem straightforward conceptually, its execution required careful design and innovation to achieve the reported gains. For instance, Cascaded Diffusion Models are conceptually simple, yet their execution is far from trivial, requiring meticulous adaptation to work effectively. Similarly, ToddlerDiffusion balances simplicity with the complexity of execution to deliver improvements in multiple dimensions:
>     * Interpretability and Interactive Generation: By introducing intermediate stages like sketches and palettes, we offer users fine-grained control and insights into the generation process.
>     * Faster Convergence: The modular design simplifies learning tasks, significantly reducing training time.
>     * Consistency in Editing: The staged approach ensures better controllability and consistency when modifying specific parts of the generated outputs.
> 2. **[Future Potential of ToddlerDiffusion]:**
> Beyond the current scope, ToddlerDiffusion paves the way for exciting future applications, some of which we have already started exploring:
>
>     * [Distill the knowledge from substage features for perception tasks] The intermediate outputs from Toddler Diffusion (e.g., the sketch stage) can be reused as strong geometric priors for perception tasks by distilling the knowledge into perception networks to enforce geometric or perceptual constraints, making them structure-aware and geometry-guided aligned.
>
>     * [Reusing Diffusion Features as High- and Low-Frequency Representations] Toddler Diffusion's staged outputs inherently capture distinct frequency information, with the sketch stage emphasizing low-frequency global structures and geometry and the palette and RGB stages capturing high-frequency details like textures. Integrating this knowledge into VLMs can improve performance and generalization for perception tasks.
>
>     * [Addressing Limitations in T2I Models] A significant challenge for text-to-image (T2I) models lies in handling complex prompts involving counting or spatial relations. Fixing these issues in the sketch or abstract stage will be easier and simpler due to its lightweight. In addition, the subsequent stages (e.g., sketch-to-RGB) will remain unaffected and reusable. This approach allows targeted improvements without retraining/finetuning the entire pipeline, significantly reducing computational costs.
>     * [Video Generation in Abstract Domains] Generating videos directly in the RGB domain is computationally expensive and prone to inconsistencies. Toddler Diffusion can split the problem into geometry consistency and stylistic consistency. Specifically, it generates videos in an abstract domain (e.g., sketches), ensuring geometric consistency across frames. Then, these sketches can be refined into RGB frames using another stage, Sketch-to-RGB, conditioned on a reference frame for color and style consistency.
>     * [Spatial Decompositionality] An intriguing future direction is to adapt Toddler Diffusion to cascade generation across spatial dimensions instead of modalities, e.g., Object-Level Cascading and Part-Level Cascading.
>     * [Modular Prompt Handling for T2I Models] Toddler Diffusion can simplify complex text prompts by decomposing them into modular components mapped to specific stages. For example, a detailed prompt can be split into geometry-focused instructions for the sketch stage and color-specific details for the sketch-to-palette stage.
>
> ```
> For instance:
> * Complex Prompt:
> "A bustling city street during sunset with a red double-decker bus on the right, a blue car parked on the left, people walking on the sidewalk, and sunlight casting long shadows on the ground."
> will be split into:
>
> * Structure-Focused Prompt:
> "A city street with a double-decker bus on the right, a car parked on the left, and people walking on the sidewalk."
>
> * Palette-Focused Prompt:
> "The double-decker bus is red, the car is blue, the sky shows a sunset with shadows on the ground."
> ```
>
> We hope these aspects demonstrate the originality and scalability of ToddlerDiffusion, as well as its potential to inspire further advancements in the community. We have added a more detailed discussion about the future potential in Section 4.
>
> Finally, we will incorporate your suggestion to discuss works like I2SB in the related work section, as we agree it is important to provide comprehensive context.
>
> Thank you once again for your feedback and encouragement—it means a lot to us!

---

### Official Review · Reviewer_xMYw · 2024-11-04

**Soundness:** 3
**Presentation:** 2
**Contribution:** 2
**Rating:** 6
**Confidence:** 5

**Summary:**

This work proposes ToddlerDiffusion, a cascaded sequence of relatively small diffusion models, to gradually generate different aspects of images. Those different diffusion models are connected by the Schrödinger Bridge to improve conditioning consistency.

**Strengths:**

1. **The proposed cascaded framework is reasonable and effective**. With fewer network parameters and training time, ToddlerDiffusion can outperform the standard LDM in both generation quality and conditioning consistency.

2. **Thorough experiments and ablation studies are conducted** in the main content and appendix to demonstrate the versatility of the proposed framework and validate the effectiveness of several design choices.

**Weaknesses:**

1. **Presentation of this work requires thorough improvements.**.
- The authors should use `\citep` for most cases in the manuscript, which places the authors' names and the year in parentheses. The current use of `\cite` makes the manuscript cluttered and difficult to read.
- The manuscripts contain many incoherent parts. For instance, in Lines 474-479, the authors state: "we used x_0 ... and employed LoRA fine-tuning for efficiency. **But** these two variants are crucial to make it work". The use of the transitional word "but" in this context is confusing. Later, the authors assert that "we **must** fine-tune the entire model instead of using LoRA". Then why is LoRA fine-tuning described as crucial if it is not the recommended approach?
- Notations are inconsistent in the manuscript. For example, in Equation (2), $x_t=\alpha_t x_0^i+(1-\alpha_t)y^i+\sigma_t^2 \epsilon_t$, whereas in Equation (6)~(16), $x_t=(1-\alpha_t) x_0+\alpha_t y+\sigma_t \epsilon_t$.
- What is the meaning of "$:\sigma_t^2=\alpha_t-\alpha_t^2$" in Equation (2)?

2. **Many statements in this work are incorrect or overclaimed**.
- In the abstract: "surpassing Stable-Diffusion (LDM) performance". However, all experiments conducted in the main content are compared with the LDM trained on small datasets, which are NOT Stable Diffusion (v1/v2/v3).
- The training objective of LDM or Stable Diffusion is the noise $\epsilon$. However, $\epsilon$ is NOT $x_t-x_0$ in LDM as stated in Line 266 and Table 4.
- In Line 507-509, by the SNR definition in this work, SNR does not tend to 0 as $t \rightarrow T$, as the concatenated condition $y$ also provides some signals to a certain extent similar to the proposed framework even at $t=T$. SNR should NOT be the reason that distinguishes between different conditioning methods (Concatenation v.s. Schrödinger Bridge).

**Questions:**

- Could you please provide more details about how to incorporate ToddlerDiffusion into Stable Diffusion? For instance, what text prompts are used? Additionally, is Stable Diffusion fine-tuned on CelebHQ datasets for five epochs using x_0 prediction and the Schrödinger Bridge conditioning mechanism similar to ToddlerDiffusion? Is the network architecture of Stable Diffusion modified to include sketch conditions, as demonstrated in Figure 12? If so, then how are the parameters initialized in the modified components?

- Why do the numbers of LDM parameters differ between Table 2 (263M) and Figure 13 (187M)?

- Why are the schedulers for $\alpha$ and $\sigma$ different in Figure 12 (left) and (right)? Why does $\alpha$ increase from $T$ to $0$ in the left block, but decrease from $T$ to $0$ in the right block? What does the presented $y$ represent in the first stage (Figure 12 left block) that unconditionally generates the sketch?

- Why is SB+Concat not utilized in ToddlerDiffusion, while only SB is used, despite the former demonstrating superior performance in Table 3?

---

> ### Author Response · Authors · 2024-11-22
> **Response 1, Part 1**
>
> ---
> > **Q1: More details about finetuning SD, in Lines 474-479:**
>
> In this experiment, we aim to demonstrate the seamless integration of our method into existing Stable Diffusion (SD) models, enabling conditional generation without introducing new parameters or branches, such as those required by ControlNet.
>
> To achieve this, we initially adopted $x_0$ as the training objective instead of the original SD objective, $\epsilon$, and employed Low-Rank Adaptation (LoRA) for fine-tuning. However, we observed that this approach does not converge easily due to the significant disparity between the pre-trained SD model’s training objective and the new objective. Consequently, we opted to fine-tune the entire model. Remarkably, after just 10k iterations, the model generated meaningful images that aligned perfectly with the input conditions, such as sketches or palettes.
>
> Thus, as stated in Lines 474–479, the choice of fine-tuning mechanism—whether using LoRA or fine-tuning the entire model—is a critical design decision for the successful implementation of our approach.
>
> We have updated this part to make it clearer for readers.
>
> Additionally, to evaluate the model under more challenging and diverse conditions, we fine-tuned SD on the Lion-ART dataset. Specifically, we randomly sampled 1M images from Lion-ART and trained the model for a single epoch, starting from SDv1.5 weights. As illustrated in Figure 24, our model effectively generates RGB images that adhere to both the input conditions: the text prompt and the sketch.
>
> ---
> > **Q2: Inconsistency between Eq2 and Eq6-16:**
>
> Thanks for pointing out this typo. In the revised version, we changed Eq2 to align with the derivation's notation in Appendix A.1.
>
> ---
> > **Q3: What is the meaning of $\sigma$ part in Equation (2)?**
>
> In Equation (2), $\sigma^2_t$ represents the uncertainty associated with the transition between the two domains. This follows a spherical formulation designed to construct a probabilistic bridge between the distributions $\rho_T$ and $\rho_0$.
>
> At the extremes of the bridge—corresponding to the endpoints of the two domains—we are fully confident in being within one of the distributions (either $\rho_T$ or $\rho_0$). Consequently, $\sigma^2_t = 0$ at these edges.
>
> However, as we move away from these edges and approach the midpoint of the bridge, the uncertainty increases, reaching its maximum value. This behavior forms the "bridge" shape, where $\sigma^2_t$ serves as a measure of uncertainty during the interpolation between the two domains.
>
> ---
> > **Q4: The training objective of LDM is $\epsilon$, which is NOT $x_t-x_0$:**
>
> Thank you for pointing out this distinction. We acknowledge that $\epsilon$ is not equal to $x_t - x_0$ but rather proportional to it:
>
> $\epsilon \propto (x_t-x_0)$
>
> The key message in Lines 264–266 is to emphasize that, in the case of Latent Diffusion Models (LDM), for an arbitrary training step $t$, predicting the difference ($x_t - x_0$) can be simpler than directly predicting the original $x_0$.
>
> However, in our case, the difference takes the form:
>
> $x_t-x_0 = \alpha_t (y-x_0) + \sigma_t \epsilon_t$
>
> This formulation introduces an additional term involving $y$, making the prediction task more complex than directly predicting $x_0$. The added complexity stems from the interplay between the conditioning variable $y$ and the uncertainty term $\sigma_t \epsilon_t$.
>
> ---
> > **Q5: In Lines 507-509, SNR should NOT be the reason that distinguishes between different conditioning methods, as the concatenated condition also provides some signals to a certain extent similar to the proposed framework even at t-->T:**
>
> Thank you for raising this important point. In Lines 215–217, we analyze the Signal-to-Noise Ratio (SNR) in the context of the unconditional setup and compare our approach against Latent Diffusion Models (LDM).
>
> Even in the conditional setup, our method demonstrates a superior SNR. Specifically, during sampling, the starting signal at
> $t=T$ in the concatenation-based conditioning framework is a mix of pure noise and the conditioning signal $y$.
> This mixture results in a very low SNR.
>
> In contrast, in our proposed framework, $x_T=y$ (not Gaussian noise).
> Since $y$ is highly relevant to the final output $x_0$ (e.g., a sketch or palette), our SNR at $t=T$ is significantly better.
> This advantage leads to improved conditioning and alignment with the target output.
>
> Additionally, we highlight that our method is flexible and can incorporate both conditioning mechanisms: concatenation and Schrödinger Bridge. As shown in Table 3, combining these approaches yields the best results, leveraging the strengths of both frameworks.

---

> ### Author Response · Authors · 2024-11-22
> **Response 1, Part 2**
>
> ---
> > **Q6: Could you please provide more details about how to incorporate ToddlerDiffusion into Stable Diffusion?**
>
> > **Q6.1: Is Stable Diffusion fine-tuned on CelebHQ datasets for five epochs using $x_0$ prediction and the Schrödinger Bridge conditioning mechanism similar to ToddlerDiffusion?**
>
> Yes, precisely. We started with the SDv1.5 weights and fine-tuned the entire model for five epochs on the CelebHQ dataset, which consists of approximately 30k images. During this process, we employed the Schrödinger Bridge conditioning mechanism (Equation 2), where $y$ represents the sketch and $x_0$ is the RGB image.
>
> > **Q6.2: Is the network architecture of Stable Diffusion modified to include sketch conditions, as demonstrated in Figure 12? If so, then how are the parameters initialized in the modified components?**
>
> No, we did not modify any part of the architecture. This is a key advantage of our approach: it seamlessly incorporates conditional abilities, such as sketch conditioning, into an existing model (e.g., SDv1.5) without introducing new parameters, unlike methods such as ControlNet.
>
> Specifically, the architecture—including the VQ-GAN encoder-decoder and the U-Net backbone—remains entirely unchanged, allowing us to directly load the pre-trained SD weights without any architectural adjustments.
>
> Formally, we made two modifications to adapt the model for our framework:
>
> 1. **Forward Process**: We replaced the standard forward process with the formulation defined by our Schrödinger Bridge approach (Equation 2).
>
> 2. **Training Objective**: We transitioned from the conventional
> ϵ-based objective to an $x_0$-based objective.
>
> As demonstrated in the table below, our method achieves superior quality to ControlNet after only five epochs of fine-tuning, showcasing the efficiency and effectiveness of our approach.
>
> | **Method** | **Unet Parameters** | **Training Time** | **FID** |
> |------------|---------------------|-------------------|---------|
> | **ControlNet** | 1.22 B              | **1.2 hr**            | 13.6    |
> | **Ours**       | **859 M**               | 1.3 hr            | **9.2**     |
>
>
> > **Q6.3: What text prompts are used?**
>
> For the initial experiment, we used a fixed text prompt across all samples:
> ``High-quality human face.''
> The goal of this experiment was to demonstrate the seamless addition of conditional capabilities to existing SD models without requiring changes to the architecture.
>
> **[Text Conditioning on CelebHQ]**
> Inspired by this discussion, we conducted additional experiments to condition the model on both text and sketch, as described in Appendix B.6. For this purpose, we extended the CelebHQ dataset by adding detailed text descriptions for each image, generated using Llava-Next v1.6. Since the descriptions produced by Llava were excessively long, we employed GPT-4o to summarize them into concise descriptions.
>
> We fine-tuned the model starting from the SDv1.5 weights for five epochs. As illustrated in Figure 23 of Appendix B.6, we provided the same sketch with different text prompts that varied only by gender. The model successfully generated RGB images that followed both the sketch and the text conditioning, showcasing its ability to integrate both modalities effectively.
>
> **[LionArt]**
> To further evaluate the robustness of our approach, we conducted experiments on a more challenging and diverse dataset, LionArt, conditioning the model on both text and sketch. We randomly sampled 1M images from LionArt and fine-tuned the model for one epoch, starting from SDv1.5 weights.
> As shown in Figure 24, our model consistently generated RGB images that adhered to both the text prompt and the input sketch, demonstrating its effectiveness even in complex scenarios.
>
> ---
> > **Q7: Why do the numbers of LDM parameters differ between Table 2 (263M) and Figure 14 (187M)?**
>
> The difference arises because Table 2 represents the full architecture's complexity, while Figure 14 focuses exclusively on the first stage ($1^{st}$ stage) of the process (Sketch to RGB).
>
> In more detail:
>
> * Table 2: This provides a comprehensive analysis of the CelebHQ dataset, comparing the complexity of all stages in our method versus standard LDM. The parameter count of 263M corresponds to the standard official network size used in the LDM source code for training on CelebHQ, where the model generates RGB images starting from noise.
>
> * Figure 14: Here, we specifically compare the first stage of the process, which generates a sketch from noise. Since the sketch modality is simpler, we employ a smaller U-Net architecture for LDM with 187M parameters.
>
> It is important to note that, as shown in Figure 13, our approach achieves much higher accuracy than LDM, even while using an architecture that is 98x smaller (1.9M parameters). This highlights the efficiency and effectiveness of our method.

---

> ### Author Response · Authors · 2024-11-22
> **Response 1, Part 3**
>
> ---
> > **Q8: Why are the schedulers for alpha and sigma different in Figure 12 (left) and (right)?**
>
> **[For $\sigma$:]**
>
> As described earlier, $\sigma$ represents the uncertainties.
> In the second and third stages (Figure 12, right), where we have a meaningful condition $y$ (e.g., a sketch or palette), the uncertainties follow the Schrödinger Bridge formulation. At the edges of the bridge, we are fully confident in being within one of the distributions ($\rho_T$ for the sketch/palette and $\rho_0$ for RGB), resulting in $\sigma^2_t=0$.
>
> In contrast, in the first stage (Figure 12, left), we start from a black canvas with added noise. This can be interpreted as adding brighter points to the canvas to act as control points during the progressive denoising steps. Consequently, $\sigma$'s behavior differs between these stages to reflect the nature of the task.
>
> **[For $\alpha$:]**
>
> $\alpha$ controls the linear interpolation between the condition $y$ and $x_0$.
> In the left part of Figure 12, there was an error in the depiction of $\alpha$.
> In the updated version, we have corrected this, ensuring that $\alpha$ remains consistent across all stages.
>
> ---
> > **Q9: What does the presented $y$ represent in the first stage (Figure 12 left block) that unconditionally generates the sketch?**
>
> In the first stage (Figure 12, left), we start from a black canvas with random added noise. This can be interpreted as adding brighter points to the canvas to act as control points during the progressive denoising steps.
>
> ----
> > **Q10: Why is SB+Concat not utilized in ToddlerDiffusion, while only SB is used, despite the former demonstrating superior performance in Table 3?**
>
> While SB+Concat shows superior performance in Table 3 for conditional generation—when the input condition $y$ is the ground truth (GT)—its performance diminishes in the unconditional setup.
>
> In the unconditional setup, $y$ is generated from the previous stage rather than being a perfect GT. This introduces errors into the generated $y$, leading to error accumulation, as discussed in Appendix A.7.
> Consequently, the model's performance suffers when relying on concatenation for conditioning.
> This is intuitive as $y$ is fed to the network in each denoising step $t$. Thus, the network emphasizes a lot on the $y$; therefore, it is more vulnerable to error accumulation.
>
> To address this, we use only the Schrödinger Bridge (SB) as the conditioning mechanism for unconditional generation because of its demonstrated robustness against accumulated errors.
>
> **[New Experiment]**
>
> Inspired by this discussion, we tested a compromise approach by applying concatenation for 50% of the steps and then dropping the condition from concatenation by feeding zeros.
> The experiment is conducted on the LSUN-Churches dataset.
> The results in the table below indicate that this trick improves performance compared to using concatenation in all steps. However, it still performs worse than using SB alone on generated sketches.
>
> | **Generated Sketch (Not GT)** | **FID** |
> |-------------------------------|---------|
> | SB                            | **6.6** |
> | SB+Concat (100% steps)        | 10.1    |
> | SB+Concat (50% steps)         | 8.9    |
>
> **Conclusion:**
>
> The choice of y-feeding strategy depends on the setup:
>
> * Unconditional generation: Use only the Schrödinger Bridge, as it exhibits superior robustness against noise in the input condition.
>
> * Conditional generation: Use a combination of Concatenation and Schrödinger Bridge, as shown in Table 3, for optimal results.
>
> We have added this nice discussion to Appendix A.8 in the revised version of the paper.
>
> ---
> > **Q11: use \citep:**
>
> We appreciate the reviewer's suggestion and have incorporated it in the revised version.

---

> ### Author Response · Authors · 2024-11-29
> **Kind reminder: We are looking forward to your reply**
>
> Dear Reviewer xMYw,
>
> We kindly ask if our response has addressed your concerns.
> Fortunately, we still have till December 3rd to discuss. Therefore, please feel free to share any additional questions or feedback, and we’ll be happy to provide further clarification.
>
> Best regards,
> The Authors

---

> > ### Comment · Reviewer_xMYw · 2024-11-30
> >
> > Dear authors,
> >
> > I apologize for the late response. I have reviewed the revised manuscript and the feedback from the other reviewers. Most of my concerns have been addressed. While I still have some concerns about the novelty, I appreciate the extensive replies provided and the additional experiments conducted. As a result, I have decided to raise my score to 6 to reflect this.

---

> > > ### Author Response · Authors · 2024-12-01
> > > **Thanks and Final Comments on the Novelty**
> > >
> > > We sincerely thank you for your thoughtful feedback and for raising the score.
> > >
> > > While we understand that novelty can be subjective, we appreciate your recognition of the value of our method to the field.
> > > As a final effort to convince you about our novelty, we would like to share two key points:
> > >
> > > 1. **[Straightforward Does Not Diminish Originality]:**
> > > While ToddlerDiffusion may seem straightforward conceptually, its execution required careful design and innovation to achieve the reported gains. For instance, Cascaded Diffusion Models are conceptually simple, yet their execution is far from trivial, requiring meticulous adaptation to work effectively. Similarly, ToddlerDiffusion balances simplicity with the complexity of execution to deliver improvements in multiple dimensions:
> > >     * Interpretability and Interactive Generation: By introducing intermediate stages like sketches and palettes, we offer users fine-grained control and insights into the generation process.
> > >     * Faster Convergence: The modular design simplifies learning tasks, significantly reducing training time.
> > >     * Consistency in Editing: The staged approach ensures better controllability and consistency when modifying specific parts of the generated outputs.
> > > 2. **[Future Potential of ToddlerDiffusion]:**
> > > Beyond the current scope, ToddlerDiffusion paves the way for exciting future applications, some of which we have already started exploring:
> > >
> > >     * [Distill the knowledge from substage features for perception tasks] The intermediate outputs from Toddler Diffusion (e.g., the sketch stage) can be reused as strong geometric priors for perception tasks by distilling the knowledge into perception networks to enforce geometric or perceptual constraints, making them structure-aware and geometry-guided aligned.
> > >
> > >     * [Reusing Diffusion Features as High- and Low-Frequency Representations] Toddler Diffusion's staged outputs inherently capture distinct frequency information, with the sketch stage emphasizing low-frequency global structures and geometry and the palette and RGB stages capturing high-frequency details like textures. Integrating this knowledge into VLMs can improve performance and generalization for perception tasks.
> > >
> > >     * [Addressing Limitations in T2I Models] A significant challenge for text-to-image (T2I) models lies in handling complex prompts involving counting or spatial relations. Fixing these issues in the sketch or abstract stage will be easier and simpler due to its lightweight. In addition, the subsequent stages (e.g., sketch-to-RGB) will remain unaffected and reusable. This approach allows targeted improvements without retraining/finetuning the entire pipeline, significantly reducing computational costs.
> > >     * [Video Generation in Abstract Domains] Generating videos directly in the RGB domain is computationally expensive and prone to inconsistencies. Toddler Diffusion can split the problem into geometry consistency and stylistic consistency. Specifically, it generates videos in an abstract domain (e.g., sketches), ensuring geometric consistency across frames. Then, these sketches can be refined into RGB frames using another stage, Sketch-to-RGB, conditioned on a reference frame for color and style consistency.
> > >     * [Spatial Decompositionality] An intriguing future direction is to adapt Toddler Diffusion to cascade generation across spatial dimensions instead of modalities, e.g., Object-Level Cascading and Part-Level Cascading.
> > >     * [Modular Prompt Handling for T2I Models] Toddler Diffusion can simplify complex text prompts by decomposing them into modular components mapped to specific stages. For example, a detailed prompt can be split into geometry-focused instructions for the sketch stage and color-specific details for the sketch-to-palette stage.
> > >
> > > ```
> > > For instance:
> > > * Complex Prompt:
> > > "A bustling city street during sunset with a red double-decker bus on the right, a blue car parked on the left, people walking on the sidewalk, and sunlight casting long shadows on the ground."
> > > will be split into:
> > >
> > > * Structure-Focused Prompt:
> > > "A city street with a double-decker bus on the right, a car parked on the left, and people walking on the sidewalk."
> > >
> > > * Palette-Focused Prompt:
> > > "The double-decker bus is red, the car is blue, the sky shows a sunset with shadows on the ground."
> > > ```
> > >
> > > We hope these aspects demonstrate the originality and scalability of ToddlerDiffusion, as well as its potential to inspire further advancements in the community. We have added a more detailed discussion about the future potential in Section 4.

---

### Author Response · Authors · 2024-11-22
**General Response 1**

We sincerely thank the reviewers for their thoughtful feedback and constructive comments. Your insights have significantly helped us strengthen our work, improve clarity, and provide additional evidence to support our claims.

We have incorporated all feedback into the revised manuscript, with changes highlighted in blue for your convenience.

Below, we summarize the key updates and experiments added to address your concerns:

1. **Handling Complex Conditions (Text):**

* Demonstrated the capability of our approach to handle complex conditions, such as text prompts.

* Conducted experiments on Lion-Art, CUB200, and Dress Code datasets.

* Qualitative results are provided in Figures 23, 24, 25, and 26.

2. **Integration with SDv1.5:**

* Integrated our approach into SDv1.5 and demonstrated its ability to generate high-quality images using limited training iterations.

* New conditioning capabilities, including sketch conditioning, are showcased in Figures 23 and 24.

3. **Conditioning Mechanism Error Vulnerability:**

* Performed a comparison between concatenation and Schrödinger Bridge, in terms of the error vulnerability (Table 9).

4. **Editing Capability Assessment:**

* Performed a human evaluation to quantitatively assess editing capabilities compared to SEED-X, providing robust evidence of our method’s superiority (Table 8).

4. **Corrections and Clarifications:**

* Fixed typos and added clarifications to improve readability and address specific points raised by reviewers.


We hope these updates comprehensively address your concerns and provide further confidence in our contributions. Thank you again for your invaluable feedback and for helping us improve our work.

---

### Meta-Review · Area_Chair_4ybA · 2024-12-23

**Metareview:**

The submission proposes ToddlerDiffusion, a cascaded sequence of diffusion models operating on different modalities such as contours, palettes, and textures, to finally generate an RGB image.
Instead of using concatenation of different modalities as conditioning, they use the Schrödinger Bridge for consistency.
The submission received final ratings of 6, 6, 8, 6.
The ACs did not find enough reason to overturn the positive consensus and recommend acceptance.

**Additional Comments On Reviewer Discussion:**

The reviewers requested clarifications on the writing and missing details/comparisons to baselines. The authors sufficiently addressed these concerns, and the reviewers updated their scores, making them more positive.

---

### Decision · Program_Chairs · 2025-01-22

Accept (Poster)